# A conserved role for the ALS-linked splicing factor SFPQ in repression of pathogenic cryptic last exons

Patricia M. Gordon[1,2 ✉], Fursham Hamid [1,2], Eugene V. Makeyev[1] & Corinne Houart[1 ✉]

The RNA-binding protein SFPQ plays an important role in neuronal development and has been associated with several neurodegenerative disorders, including amyotrophic lateral sclerosis (ALS), frontotemporal dementia (FTD), and Alzheimer's disease. Here, we report that loss of *sfpq* leads to premature termination of multiple transcripts due to widespread activation of previously unannotated cryptic last exons (CLEs). These SFPQ-inhibited CLEs appear preferentially in long introns of genes with neuronal functions and can dampen gene expression outputs and/or give rise to short peptides interfering with the normal gene functions. We show that one such peptide encoded by the CLE-containing *epha4b* mRNA isoform is responsible for neurodevelopmental defects in the *sfpq* mutant. The uncovered CLE-repressive activity of SFPQ is conserved in mouse and human, and SFPQ-inhibited CLEs are found expressed across ALS iPSC-derived neurons. These results greatly expand our understanding of SFPQ function and uncover a gene regulation mechanism with wide relevance to human neuropathologies.

[1] Centre for Developmental Neurobiology and MRC Centre for Neurodevelopmental Disorders, IoPPN, Guy's Campus, King's College London, London SE1 1UL, UK. [2]These authors contributed equally: Patricia M. Gordon, Fursham Hamid. ✉email: patricia.gordon@kcl.ac.uk; corinne.houart@kcl.ac.uk

Neurons are highly polarized cells with specialized compartments that must be able to respond to growth cues as well as to form and modify their synapses in an activity-dependent manner. Each compartment of a neuron is able to achieve functional specificity by maintaining a unique transcriptome and proteome[1–4]. This points to the importance of post-transcriptional regulation of gene expression in this biological context. Indeed, RNAs from neuronal tissue are regulated by a complex array of alternative splicing, intron retention, telescripting, recursive splicing, and alternative cleavage and polyadenylation events[5–16].

Splicing Factor Proline/Glutamine Rich (SFPQ) is a ubiquitously expressed RNA-binding protein of the DBHS family with diverse roles in alternative splicing, transcriptional regulation, microRNA targeting, paraspeckle formation, and RNA transport into axons[17–24]. Inactivation of the *sfpq* gene causes early embryonic lethality in mouse and zebrafish as well as impaired cerebral cortex development, reduced brain boundary formation, and axon outgrowth defects[23,25–27]. In humans, *sfpq* mutations have been linked to neurodegenerative diseases such as Alzheimer's, ALS, and FTD, and SFPQ interacts with the ALS-associated RNA-binding proteins TDP-43 and FUS[28–33].

While SFPQ is known to play a role in alternative splicing, only a few RNA targets of SFPQ have been identified. Intriguingly, SFPQ has opposing effects on splicing, depending on the target: it represses inclusion of exon 10 of *tau* and exon 4 of *CD45*, but conversely it promotes inclusion of the N30 exon of non-muscle myosin heavy-chain II-B[19,30,34–36]. In addition to its role in splicing, SFPQ has been shown to be part of the 3′-end processing complex, where it enhances cleavage and polyadenylation at suboptimal polyadenylation sites[37–39]. The mechanisms by which SFPQ regulates mRNA processing are still unclear, however, and more work is necessary to understand its contribution to normal and pathological cell states.

To understand the molecular functions of SFPQ in developing neurons, we performed an RNA-seq analysis of *sfpq* homozygous null mutant and sibling zebrafish embryos at 24 hpf, the stage of phenotypic onset, and undertook a multi-level comparative analysis. Our results reveal an important role for the protein: loss of SFPQ causes premature termination of transcription as a result of previously unannotated pre-mRNA processing events that we refer to as Cryptic Last Exons (CLEs). Here we describe the formation of CLEs and show that not only do the truncated transcripts act as a form of negative regulation of gene expression levels, but they also directly contribute to the *sfpq* pathology. This function of SFPQ is conserved across vertebrates and is likely to be implicated in human SFPQ-mediated disease states.

## Results

### Identification of the SFPQ-dependent splicing regulation program.
To examine the effect of SFPQ on gene expression and RNA splicing, we analyzed total RNA extracted from 24 hpf *sfpq*$^{-/-}$ zebrafish embryos and their heterozygous or wild-type siblings by RNA sequencing (RNA-seq). Differential gene expression analysis using the TopHat/Cufflinks workflow[40] uncovered 189 genes that were upregulated and 1044 genes that were downregulated in the mutant samples by a factor of at least 1.3-fold with $q \leq 0.05$ (Fig. 1a). These results are consistent with our previous microarray study, which showed the vast majority of genes with differential expression in *sfpq*$^{-/-}$ embryos as being downregulated[26]. Gene ontology (GO) analysis of the new dataset, using total transcribed genes as a background gene set, showed enrichment for neuron-specific terms, including neuronal differentiation and axon guidance (Supplementary Data 1 and 2). Using Cufflinks' differential isoform switch analysis, we identified 112

genes with significant changes in the relative expressions of splice variants in the mutants ($q \leq 0.05$; Supplementary Data 3). GO analysis of these genes again showed an over-representation of neuron-specific terms including axonogenesis, axon guidance, and dendrite formation (Supplementary Data 4). Surprisingly, 52 of the 112 genes were predicted to express a splice variant containing a cryptic alternate last exon, not annotated in the current zebrafish assembly (Supplementary Fig. 1a). To verify this, we analyzed the dataset with Whippet[41], a tool that sensitively detects changes in the usage of alternative exons and quantifies gene expression changes. Whippet also uncovered a high proportion of down-regulated genes in *sfpq*$^{-/-}$ embryos (Fig. 1b). More importantly, the analysis confirmed that alternate last exons are the most abundant category of SFPQ-regulated splicing events accounting for 18.5% of the total (Fig. 1c). Systematic classification of these exons into known and "cryptic" corroborated that the majority of these events (113 out of 157) have not been previously annotated (Fig. 1d and Supplementary Data 5). These last exons were expressed from 103 genes, 25 of which were also detected by Cufflinks (Supplementary Fig. 1b). In 65 of the 103 genes, the cryptic last exons were predicted to give rise to truncated peptides with missing C-terminal domains (Supplementary Data 6). We refer to this pervasive splicing defect, in which the transcript undergoes premature termination after the inclusion of a previously unknown exon, as Cryptic Last Exons (CLEs; Fig. 1e).

### The use of CLEs inversely correlates with expression of full-length transcripts.
Of the 103 CLE-containing genes, 97% exhibited increased splicing of CLEs in *sfpq*$^{-/-}$ (Supplementary Data 5). Notably, almost half of these genes were downregulated in mutants, indicating concurrent alterations in expression level and splicing for these genes (~13-fold enrichment over the number expected by chance; Fisher's exact test $p = 4.66 \times 10^{-49}$; Fig. 1f). Indeed, we saw a significant negative correlation between CLE inclusion and overall gene expression levels (Pearson correlation $r = -0.27$, $p = 3.2 \times 10^{-3}$; Supplementary Fig. 1c). The full-length (non-CLE) isoforms showed an even stronger enrichment for the downregulation effect (exceeding the expectation ~25-fold; Fisher's exact test $p = 6.03 \times 10^{-80}$), suggesting that CLEs often dampen the production of full-length transcripts (Fig. 1g). To verify these results, we analyzed five selected CLE-containing genes, *nbeaa*, *gdf11*, *epha4b*, *trip4*, and *b4galt2*, by reverse transcription-quantitative PCR (RT-qPCR). In all cases, cryptic exons showed a substantial increase in expression level in *sfpq*$^{-/-}$ mutants compared to siblings (Supplementary Fig. 1d–h). Additionally, we detected a strong downregulation of the full-length isoforms in four of the five genes, suggesting that the loss of *sfpq* causes upregulation of the CLE isoforms at the expense of their full-length counterparts (Supplementary Fig. 1e–h). These results argue that SFPQ may maintain stable gene expression by repressing CLEs.

### CLE-terminated transcripts are cleaved and polyadenylated at the 3′ end.
The last exons included into mature mRNAs are expected to acquire a 3′-terminal poly(A) tail as a result of pre-mRNA cleavage and polyadenylation. To check if CLEs contained this feature, we analyzed poly(A)-proximal mRNA sequences by 3′ mRNA-seq and identified putative cleavage/polyadenylation positions as major clusters of 3′ mRNA-seq reads (see Methods section). Notably, 84 out of the 113 CLEs contained or were immediately followed by cleavage/polyadenylation clusters (Supplementary Fig. 2a, b). The incidence of cleavage/polyadenylation clusters was similar for the constitutive last exons (Supplementary Fig. 2b). Importantly, cleavage/polyadenylation events downstream of CLEs tended to be significantly upregulated in *sfpq*$^{-/-}$

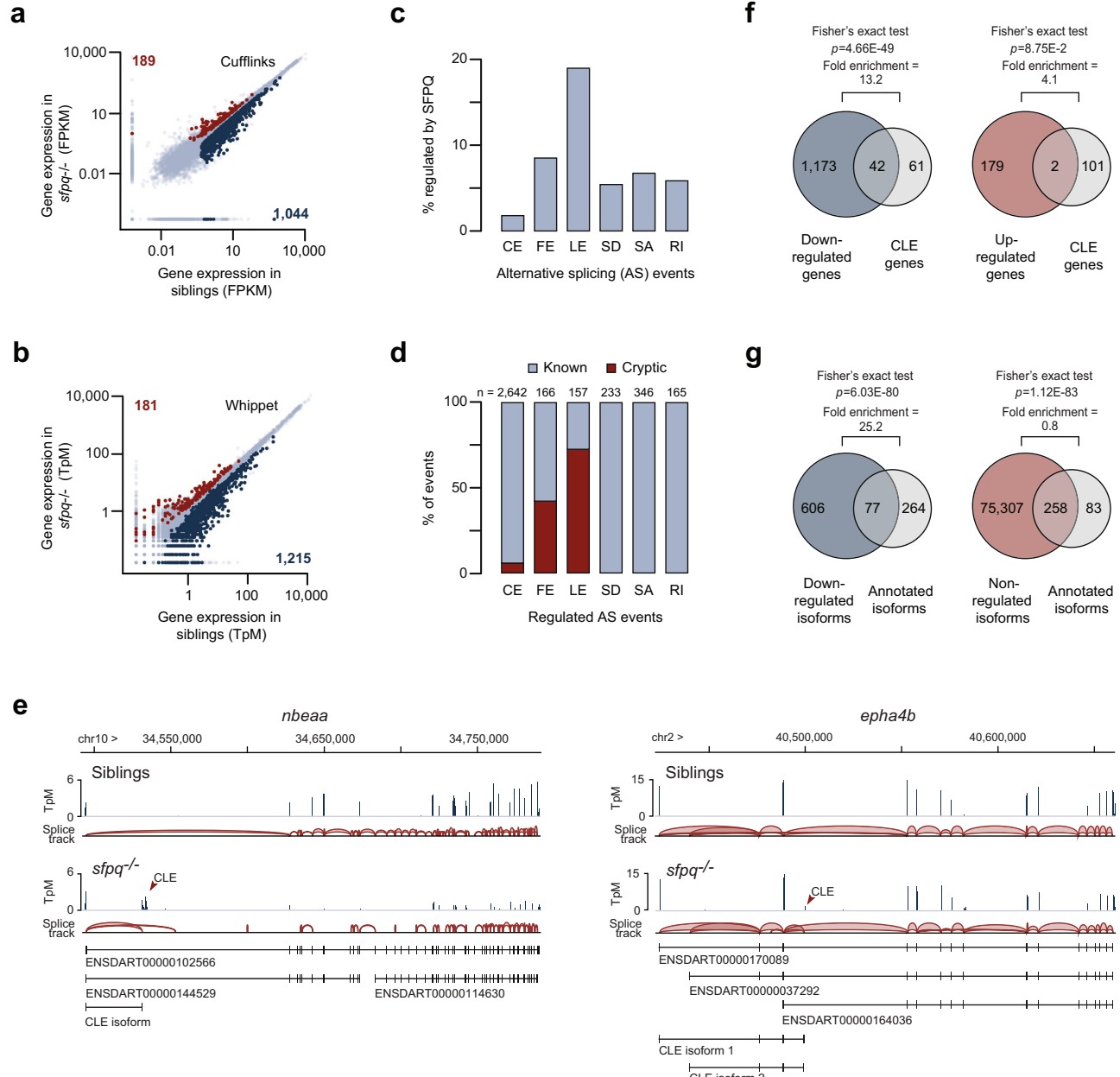

**Fig. 1 SFPQ regulates the formation of cryptic last exons (CLEs). a, b** Scatter plot showing expression values of genes in $sfpq^{-/-}$ and siblings, analyzed using Cufflinks (**a**) or Whippet (**b**) pipelines. FPKM fragments per kilobase of transcript per million, TpM transcripts per million. **c** Alternative last exons is a major category of SFPQ-controlled events. CE cassette exon, FE first exon, LE last exon, SD splice donor, SA splice acceptor, RI retained intron. **d** Majority of SFPQ-regulated last exon events are cryptic. **e** Sashimi plots showing example CLE formation in *nbeaa* and *epha4b*. Top tracks: plot of read coverage from siblings (upper) and $sfpq^{-/-}$ (lower). Bottom tracks: isoforms discovered for each gene. **f** Genes expressing CLE-containing isoforms tend to be downregulated in $sfpq^{-/-}$. Two-sided Fisher's exact test was performed. **g** Normal long isoforms (annotated isoforms) from CLE-expressing genes tend to be downregulated in $sfpq^{-/-}$. Two-sided Fisher's exact test was performed.

mutants, while the use of cleavage/polyadenylation sites in the corresponding constitutive last exons was often reduced (Fig. 2a–c and Supplementary Fig. 2c). Metaplots of the change in 3′ mRNA-seq read coverage confirmed this effect (Fig. 2d).

To validate the 3′ mRNA-seq results, we performed 3′ RACE on 10 CLE-containing transcripts: *bcas3*, *dlg5a*, *epha4b*, *gdf11*, *gli2b*, *immp2l*, *nbeaa*, *sema5ba*, *trip4*, and *vti1a* (Fig. 2e). Of the 10 CLEs, 7 had the canonical polyadenylation signal (PAS) AATAAA just upstream of the predicted cleavage site, while the remaining 3 had common one-base substitutions ATTAAA and AATATA[42]. These results indicate that normal expression of SFPQ is required to repress cryptic cleavage/polyadenylation sites.

**CLEs tend to occur in long introns and show evidence of interspecies conservation.** In order to understand under what conditions CLEs form, we examined CLE-containing introns and compared them to all other introns from the same genes. We first asked whether CLE-containing introns have a specific position within their genes and found no significant bias (Fig. 3a). However, when we ranked the introns by length, we found that CLEs are frequently located in the longest intron of the gene (Fig. 3b) and that CLE-containing introns are significantly longer than other introns in general (Fig. 3c). To test if CLEs are more commonly found in long introns due to simple probabilistic considerations, we performed a weighted random sampling of

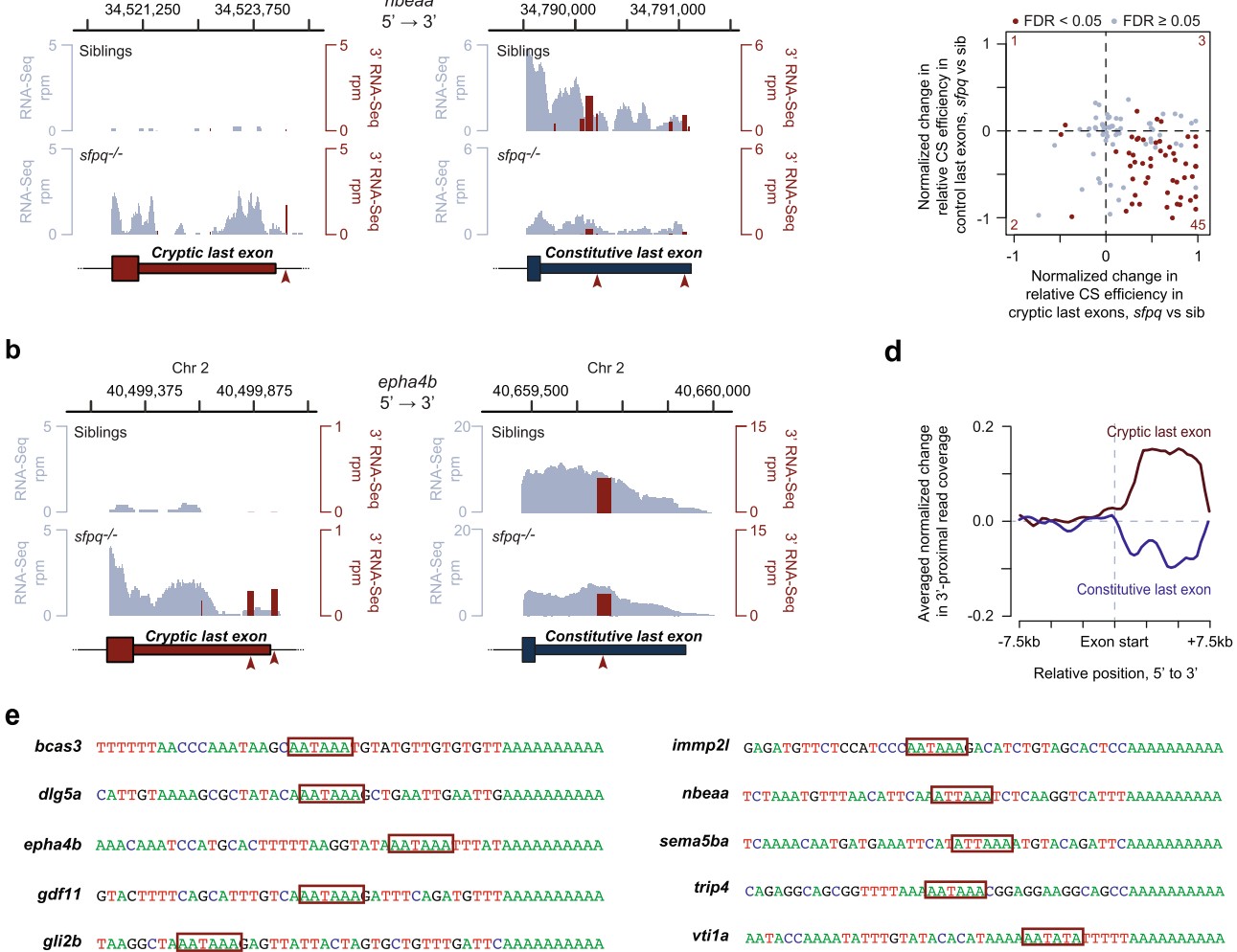

**Fig. 2 CLEs are cleaved and polyadenylated. a, b** Representative coverage plots from RNA-seq (light blue) and 3′ mRNA-seq (red) experiments showing cryptic last and constitutive last exons. Only clustered reads are shown in the 3′ mRNA-seq profile and consensus polyadenylation signals (PASs) are marked by red arrowheads. **c** Four-way dot plot representing the change in relative cleavage site usage (sfpq vs sib) for CLEs (x-axis) against its corresponding constitutive last exons (y-axis; control). A positive value on each axis represents an increased CS usage in sfpq$^{-/-}$ null. Genes showing significantly changing CS usage in both CLE and control (FDR < 0.05) are colored red and the total number of significantly regulated genes is indicated for each quadrant. **d** Metaplot of the normalized change in 3′ mRNA-seq coverage within regions surrounding CLEs (red) and constitutive last exons (blue; control). **e** Sanger sequencing of 3′RACE PCR products of CLE isoforms. PAS hexamers are shown within red boxes.

introns from CLE-containing genes setting the probability of each intron to be selected proportional to its length (Supplementary Fig. 3a). This revealed that the actual lengths of CLE-containing introns are significantly longer than expected by chance.

Consistent with the above results, CLE-containing genes were significantly longer than most zebrafish genes (Supplementary Fig. 3b). Within the intron, location of the CLE was biased toward the 5′ end, with the median position 22.4% of the way into the intron (95% confidence interval of 18.1% to 26.7%; Fig. 3d). The distance between the CLE and the upstream exon was generally <10 kb (Fig. 3e). The 3′ splice site sequences preceding CLEs were somewhat weaker than those of constitutive last exons (CLE median: 7.9 vs control median: 9.1), but both categories contained readily discernible U2AF binding motifs (Supplementary Fig. 3c, d).

We next asked whether the sequences within and neighboring the CLEs are conserved. To this end, we calculated the mean conservation scores of 1 kb sequences (sliding window, 1 bp steps) along these CLE-containing introns, using both the PhastCons and phyloP analysis methods[43,44]. Our analyses showed that sequences containing CLEs tend to have higher

conservation scores as compared to sequences within the same intron that do not contain CLEs (Supplementary Fig. 3e). In fact, 18% of these sequences displayed a mean PhastCons score of at least 0.5 (as opposed to 12% of non-CLE sequences; Fisher's exact test: $5.71 \times 10^{-51}$). However, few CLEs showed overall phyloP scores above 0.5 (Supplementary Fig. 3f, g). Next, we calculated the mean base conservation scores of each CLE together with 250 bp flanking sequences (Fig. 3f). In the sequences with higher overall PhastCons scores, we found that conservation was the highest in the region surrounding the start of the exon (Fig. 3f). Similar observations are made measuring phyloP conservation scores (Fig. S3h, i). Of note, the 3′ splice site was highly conserved in at least 10% of candidate CLEs (Supplementary Fig. 3j). Together, these data indicate that CLEs are often found close to the 5′ ends of very long introns and that at least some of these cryptic last exons are evolutionarily conserved.

**SFPQ directly binds to sequences adjacent to CLEs.** The accumulation of CLE-terminated transcripts in *sfpq* mutants raises the

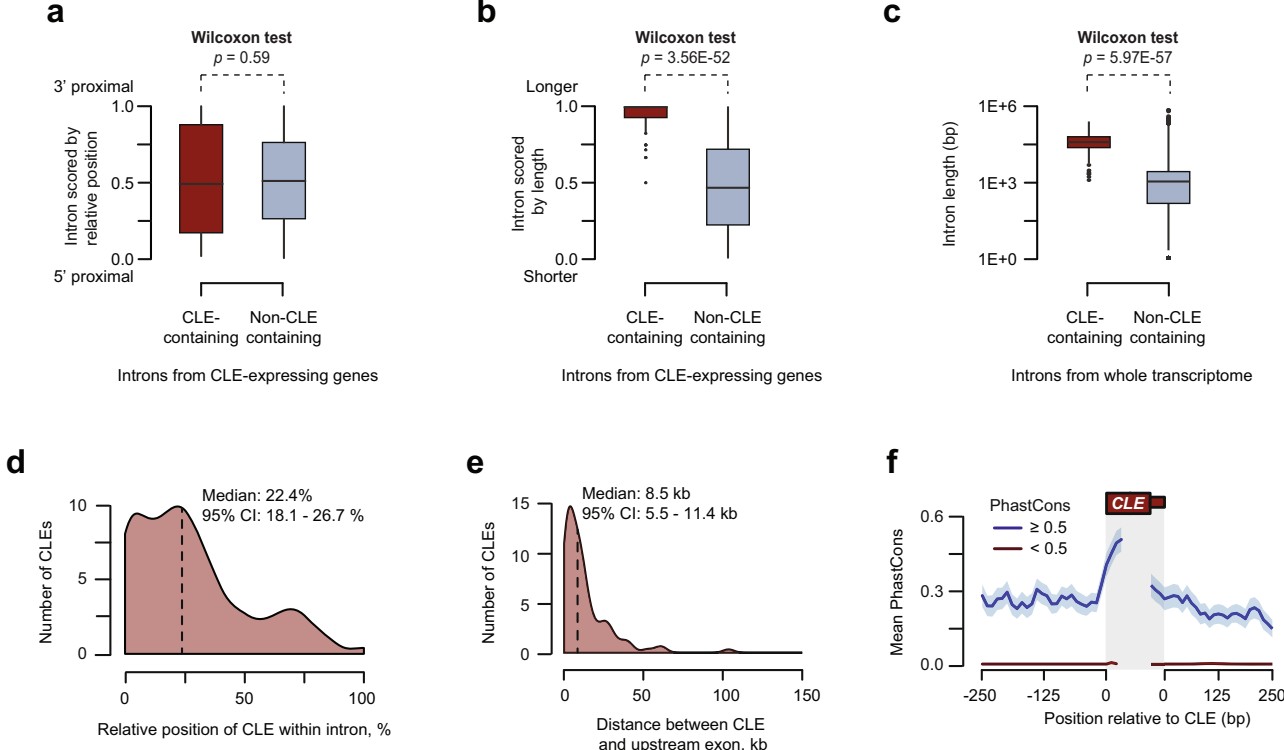

**Fig. 3 Molecular properties of CLEs. a, b** CLE containing ($n = 109$) and non-CLE ($n = 1557$) containing introns from CLE-expressing genes were scored by their relative position (**a**) and relative length (**b**), and the distributions of these scores were plotted. Note that CLE-containing introns show no gene position bias but tend to be among the longest introns in the gene. The box bounds represent the first and third quartiles and the black lines at the middle of the boxes show the medians. Top and/or bottom whiskers represent 1.5x of the range between the third and the first quartiles (interquartile range). Circles represent outliers. Two-sided Wilcoxon rank sum test was performed. **c** CLE-containing introns ($n = 109$) are longer compared to all other introns in the zebrafish transcriptome ($n = 209,012$). The box bounds represent the first and third quartiles and the black lines at the middle of the boxes show the medians. Top and/or bottom whiskers represent 1.5x of the range between the third and the first quartiles (interquartile range). Circles represent outliers. Two-sided Wilcoxon rank-sum test was performed. **d** CLEs tend to occur relatively close to the 5′ end of their host introns. **e** CLEs are found within 10 kb of the upstream constituitive exon. **f** Metaplot showing the conservation score of sequences surrounding conserved (blue) and non-conserved (red) CLEs. In all, 280 bp of surrounding intron/CLE junction sequence (250 bp intron and 30 bp exon) were binned into 10 bp windows and the mean PhastCons score for each bin is shown ±SEM.

question of whether SFPQ represses CLEs in a direct manner. SFPQ binds promiscuously to a wide range of RNA sequences[24,36], making binding prediction difficult. That said, an SFPQ-binding motif deduced in vitro[45], was significantly enriched upstream of cryptic exon sequences compared to control last exons (Fig. 4a). To verify this observation, we purified SFPQ-RNA complexes in 24 hpf embryos using standard CLIP protocol and quantified the relative amount of bound CLE RNA fragments for a handful of randomly picked CLE candidates, using RT-qPCR. Our results confirmed that SFPQ binds either within the CLE or in adjacent 5′ or 3′ intronic regions of at least three CLE transcripts (Fig. 4b–d). These results support the idea that SFPQ directly binds to region surrounding CLEs to regulate their inclusion.

**CLEs can dampen the expression of full-length transcripts.** The reciprocal relationship between CLEs and the abundance of full-length transcripts (Fig. 1f and Supplementary Fig. 1c) suggests that these exons may act as negative regulators of gene expression. If production of CLE transcripts is a mechanism for down-regulating the normal full-length transcripts, then eliminating the cryptic exon in *sfpq*−/− mutants should rescue their expression. To test this possibility, we used the gene *b4galt2* as case study, as it shows a very strong loss of expression of its three normal isoforms in the mutant (Supplementary Fig. 1g). We used

CRISPR/Cas9 to delete the *b4galt2* CLE, injecting Cas9 along with two guide RNAs that targeted directly upstream of the cryptic exon and at the 3′ end of the exon (Fig. 5a). Injected founder embryos (crispants) will show mosaicism, so a complete loss of the cryptic exon would not be expected in every cell of the embryo. With expected variation between individuals, PCR analysis of the "crispants" showed a detectable deletion band for six out of eight tested embryos (Fig. 5a). Encouraged by this result, we performed RT-qPCR on pooled injected *sfpq*−/− embryos to measure the expression levels of the normal *b4galt2* transcripts and saw a significant rescue of the longer transcripts compared to the uninjected *sfpq*−/− control, despite the natural variability in the targeting efficiency (Fig. 5a). Therefore, the *b4galt2* CLE prevents the expression of full-length transcripts. Since CRISPR/Cas9 targeting of CLEs in two other genes did not produce similar rescue effects (example *gdf11* CLE, Supplementary Fig. 5a), we concluded that CLEs regulate expression levels of normal splice variants of some but likely not all of the SFPQ-inhibited CLE-containing genes.

**Truncated protein derived from CLE-containing *epha4b* transcripts accounts for the boundary defects in *sfpq*−/− brain.** In addition to affecting the expression levels of normal isoforms, CLE transcripts could impact the *sfpq* phenotype through aberrant functions of the truncated RNAs or the short peptides they

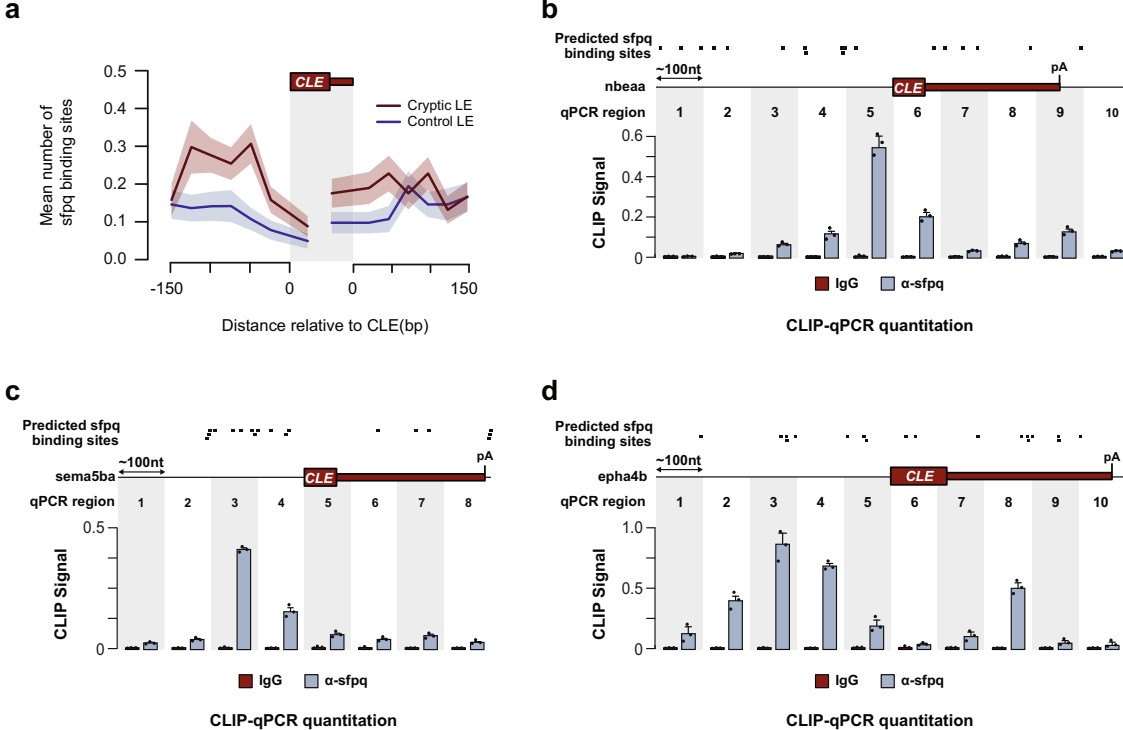

**Fig. 4 SFPQ directly binds to CLE-adjacent RNA sequences. a** Metaplot showing the distribution of predicted SFPQ-binding sites surrounding CLEs (red) and constitutive last exons of each CLE-containing gene (blue). In total, 200 bp of surrounding intron/CLE junction sequence (150 bp intron and 50 bp exon) were binned into 50 bp windows and the mean number of predicted motifs is shown ±SEM. **b**–**d** Top: location of SFPQ-binding motifs predicted using the MEME suite. Bottom: RT-qPCR quantitation showing the relative enrichment of SFPQ-interacting regions surrounding CLEs. RT-qPCR primer pairs were designed for each 100-bp sequence window demarcated by alternating gray and white areas. Abundance of SFPQ- or IgG(control)-crosslinked RNAs were normalized to input and the mean value from three replicates were shown ±SD. Source data are provided as a Source Data file.

produce. We focused on the candidate gene *epha4b*, which expresses a CLE-containing short mRNA in *sfpq* null embryos, while showing no change in expression of the normal transcripts (Supplementary Fig. 1d). This gene is one of two zebrafish paralogues of the human ALS-associated gene *Epha4*[46,47], coding for a protein-tyrosine kinase of the Ephrin receptor family known to regulate hindbrain boundary formation[48,49]. Hindbrain boundary defects are a prominent component of the *sfpq*[−/−] phenotype, and we asked whether expression of the *epha4b* CLE-containing transcript could account for this abnormality. Truncated forms of EPH receptors have been shown to act as dominant negatives by competing with full-length versions of the protein for ligand binding[50]. The predicted peptide produced by the *epha4b* CLE-containing short transcript would contain the ligand binding domain but not the transmembrane and intracellular domains and thus would be predicted to be a dominant negative (Supplementary Data 6).

To assess possible effects of the shortened *epha4b*, we first performed an in-situ hybridization using a probe for the cryptic exon. We found that in 24 hpf *sfpq*[−/−] embryos, but not in siblings, the *epha4b* CLE was expressed strongly in the midbrain and hindbrain (Fig. 5b), where the gene is normally transcribed at that developmental stage. We then tested whether, in wild-type fish, injection of the CLE transcript would induce defects in the midbrain or hindbrain. Using the early hindbrain boundary marker *rfng*, we found that injection of the short *epha4b* transcript did not affect formation of the midbrain but did cause a severe loss of hindbrain rhombomere boundaries similar to that seen in the *sfpq*[−/−] mutant (Fig. 5c). In order to distinguish whether the effect was the result of the mRNA or its translation in a truncated protein, we also injected the short *epha4b* transcript

with an early stop inserted into exon 1, prematurely terminating translation. We found that this mutated transcript did not alter hindbrain boundary formation (Fig. 5c), indicating that the boundary defect occurs as a result of expression of the truncated EphA4B protein.

We then asked whether repressing the *epha4b* CLE in *sfpq*[−/−] embryos could rescue the *sfpq* hindbrain boundary defect. We used a splice junction morpholino (MO) that targeted the 3′ splice acceptor site of the CLE to prevent the cryptic exon from being used in *sfpq* mutants. Although MOs frequently have off-target effects, those effects are generally the opposite of what we would expect to see from a rescue (i.e. increased cell death and off-target phenotypes, never rescue of phenotypes). However, as MOs have been shown to have non-specific phenotypic effects on the hindbrain[51] and also cause increases in the signal strength in in-situ hybridization, we used mismatch controls to ensure that our results were specific to the *epha4b* CLE splice-MO. We tested the MO efficiency using RT-PCR with primers both within the cryptic exon and across the exon junction (Supplementary Fig. 5b). We then examined the effects of the MO on hindbrain development using the boundary-specific *rfng* marker (Fig. 5d) and the pan-neuronal marker DeltaA (Fig. 5e). We observed that the CLE splice junction MO, but not the mismatch control, rescued formation of rhombomere boundaries in *sfpq*[−/−] mutants. Taken together, these results demonstrate that the hindbrain boundary defect in *sfpq*[−/−] embryos is driven by the dominant-negative effects of the *epha4b* CLE transcript.

**Repression of CLEs by SFPQ is conserved across vertebrates and relevant to human neuropathologies.** Given the importance

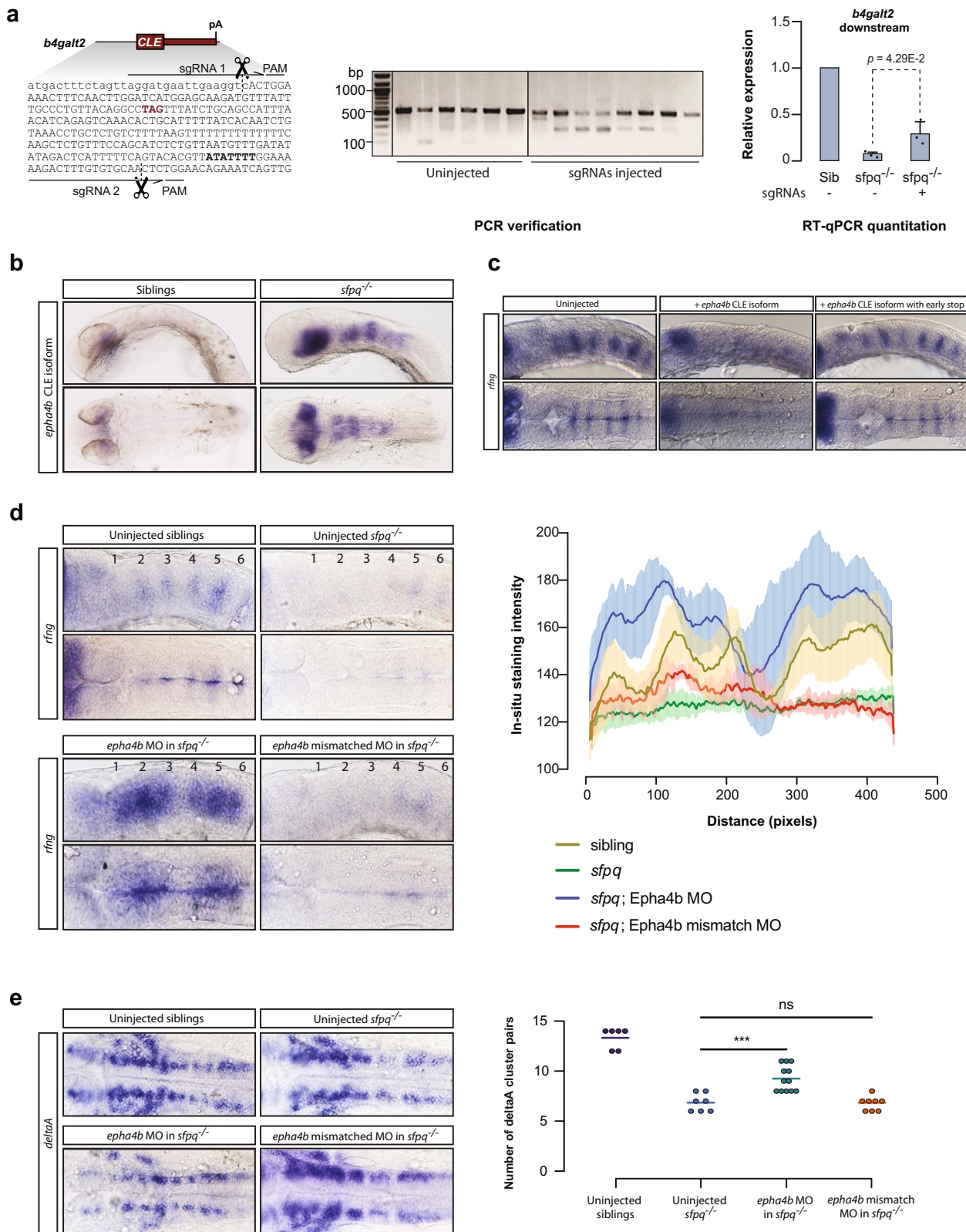

of SFPQ in human neurodegenerative diseases we explored whether SFPQ represses CLEs in mammals. We therefore analyzed publicly available RNA-seq datasets from *sfpq* loss-of-function experiments. Analyses done on different developmental stages or tissues than our RNA-seq experiment would not be expected to pick up many of the same CLEs because different groups of genes would be transcribed under those differing conditions. However, if the molecular function of SFPQ were conserved, we would still be able to identify CLE formation among the expressed genes. Using RNA-seq dataset from a mouse model with a conditional mutation inactivating *Sfpq* in the cerebral cortex[23], we identified a total of 191 Sfpq-inhibited last exons, 144 of which were cryptic (Fig. 6a and Supplementary Data 7). Importantly, the properties of CLE-containing introns in

**Fig. 5 CLEs have functional impacts. a** Deletion of the *b4galt2* CLE using CRISPR/Cas9 rescues expression of downstream exons. Left: cut sites of the *b4galt2* sgRNAs. CLE is indicated by capital letters. Center: PCR verification of Cas9 cleavage after injection of sgRNAs. Representative image; experiment performed five times. Right: RT-qPCR quantitation of the relative expression of the downstream *b4galt2* exons in *sfpq−/−* embryos compared to siblings (±SD); *n* = 3 biologically independent replicates. Two-tailed unpaired *t*-test was performed. **b** In-situ hybridization of the epha4b CLE at 24 hpf, displaying strong expression in the midbrain and hindbrain of *sfpq*−/− embryos. **c** In-situ hybridization of *rfng* shows rhombomere boundary defects at 22ss after injection into WT embryos of the *epha4b* cryptic transcript or a mutated transcript with an early stop codon. Loss of boundaries seen in 8/10 embryos. **d** Left: in-situ hybridization of *rfng* shows rhombomere boundary defects of *sfpq*−/− embryos are rescued by injection of the *epha4b* cryptic splice junction morpholino but not a mismatch morpholino. Rhombomere boundaries are numbered. Right: quantification of staining in rhombomeres in three lateral view samples for each condition. Representative images; defect seen in 13/15 embryos. **e** Left: in-situ hybridization of *DeltaA* shows a loss of discrete neuronal clusters in *sfpq*−/− which is rescued by injection of the *epha4b* cryptic splice junction morpholino but not a mismatch morpholino. Right: quantification of number of DeltaA clusters in each condition. Each data point represents one embryo; embryos examined over two independent experiments. Two-tailed *t*-test was performed, ***p* = 0.0005. **c–e** Upper: lateral view. Lower: dorsal view. Source data are provided as a Source Data file.

the mouse cortex were similar to those in zebrafish: they showed no positional bias along the gene and were often the largest in a gene and statistically larger than other mouse introns (Fig. 6b and Supplementary Fig. 6a, b). Using the CLIP-seq dataset from the same study, we found an enrichment in Sfpq-binding peaks in region surrounding CLEs but not in constitutive last exons (Fig. 6c). Among these 144 Sfpq-repressed CLEs, three orthologs of zebrafish CLE-containing genes were found: *Cpped1*, *Fam172a*, and *Exoc4* (Fig. 6d and Supplementary Fig. 6d, e). Surprisingly, the constitutive exons flanking these mouse CLEs share high sequence homology (57–77%) to the exons flanking its ortholdogous zebrafish CLEs (Fig. 6d and Supplementary Fig. 6d, e). Since low-abundance CLE-containing isoforms may escape detection by alternative splicing workflows, we manually inspected the transcriptome for other conserved CLEs. We found exonic and junction reads mapping to the beginning of intron 3 of the mouse *Epha4* gene, the closest orthologue of zebrafish *epha4b* (Fig. 6e and Supplementary Fig. 6c). Similarly, exons 3 and 4 of these orthologs share exceptionally high sequence similarity (Fig. 6e).

As SFPQ has been recently linked to ALS in human, we also examined RNA-seq results from iPSCs derived from ALS patients, which show loss of nuclear SFPQ expression[31]. In total, we found 68 CLE events upregulated in ALS-mutant backgrounds in at least one neuronal differentiation stage (Fig. 6f and Supplementary Data 8). This is probably an underestimation since the sequencing depth in this dataset was lower than that of the mouse knockout study. While most of these CLEs appear in a development-specific manner, CLEs spliced from PRPF6 and DPYSL3 genes showed consistent upregulation in three time-points (Fig. 6f, g and Supplementary Fig. 6f). The latter gene is involved in positive regulation of axon guidance and genetic variants of this gene have been previously implicated in ALS patients[52]. These results indicate that CLE repression is a conserved function of SFPQ, and that CLE-terminated transcripts may play a role in SFPQ-associated disease states.

## Discussion

Our study uncovers a critical role of SFPQ in repression of cryptic last exons (CLEs). We show that CLEs are functionally relevant as regulators of gene expression output or/and a mechanism for producing deleterious protein isoforms. Moreover, the CLE-repressing function of SFPQ is conserved in mouse and human, indicating an important developmental role, with implications for human pathology.

**Mechanism of CLE formation**. The presence of strong polyadenylation sites in CLE sequences suggests that the paucity of CLE-containing isoforms under normal physiological conditions is due to active suppression of CLE cleavage/polyadenylation or/and splicing. Our CLIP-qPCR experiments provide evidence for

SFPQ binding within or directly adjacent to CLEs. Moreover, the bias of CLEs towards the 5′ end of long introns is consistent with previous analyses of SFPQ localization on RNA[23]. These data argue that SFPQ may play a direct role in repressing cryptic exon formation. SFPQ is known to interact with Pol II, promoting Ser 2 phosphorylation, and transcriptional elongation[22,23], and our results are consistent with a mechanism in which SFPQ recruitment to CLE-containing introns directly inhibits recognition and splicing of the cryptic exon. However, further work will be required to distinguish between suppression of splicing versus blocking of the polyadenylation site.

The relationship between SFPQ and CLEs extends our understanding of the regulation possibilities afforded by long introns. Long introns have been previously shown to control gene expression through interplay between premature cleavage/polyadenylation and the U1 snRNP-dependent antitermination mechanism known as "telescripting"[14,15,53–56]. SFPQ-mediated CLE repression also operates in long introns (Fig. 3b), but the underlying mechanisms appear to be quite different. Telescripting is thought to control cleavage/polyadenylation in a splicing-independent manner, whereas the formation of CLEs is a bona fide splicing event possibly involving interactions of the U2 snRNP or/and U2AF with the cleavage/polyadenylation machinery[57]. Moreover, U1-dependent antitermination often function close to the 5′ end of the gene[55,56], whereas CLEs do not show such a gene location bias. The distinction between the two mechanisms is further highlighted by our recent RNA-seq analyses performed on embryos with a null mutation in *snRNP70*, a conserved component of the U1 snRNP[58]. We found that cryptic last exons only represent 0.3% of the total number of regulated exons in the snRNP70 null mutants, with no overlap between the rare cryptic exons found in *snrnp70* and those formed in *sfpq* mutants. These results indicate that SFPQ-mediated repression of cryptic exons is independent from telescripting.

Long introns have been also shown to be subject to recursive splicing (RS), a multistep process promoting accuracy and efficiency of intron excision[59,60]. Like CLEs, RS-sites appear primarily in long introns in genes with neuronal function. RS-sites initially produce an RS-exon that is spliced to the upstream exon prior to being excised at the subsequent round of splicing reactions. However, typical RS-exons do not contain cleavage/polyadenylation sequences, so their inclusion does not truncate the transcript. In addition, recursive splicing creates a stereotypical saw-tooth pattern of RNA-seq reads, which is not seen in the *sfpq*−/− RNA-seq data. Therefore, SFPQ and CLEs provide a distinct regulation modality compared to telescripting and recursive splicing.

**Pathology of cryptic transcripts**. A notable feature of the SFPQ-repressed CLEs is the detrimental effect that they have on the function of their host genes. Upregulation of a cryptic exon could affect gene regulation and function through both co- and post-

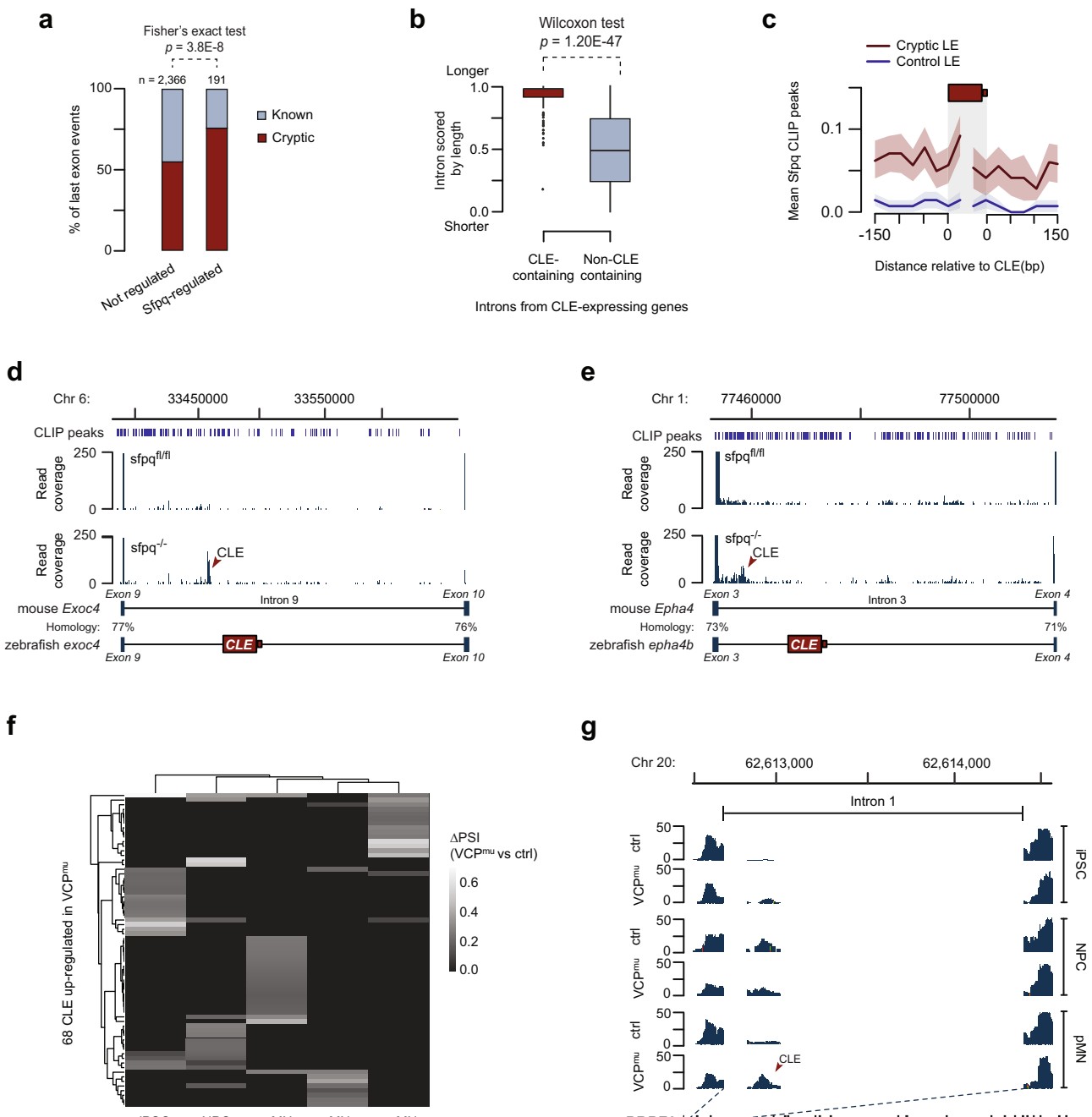

**Fig. 6 The CLE-repressing function of SFPQ is conserved in mouse and human. a** Bar plot of the proportion of unannotated (cryptic) last exons from mouse cortical neurons grouped based on their regulation by Sfpq. Two-sided Fisher's exact test was performed. **b** Introns from mouse CLE-expressing genes were scored by their relative length and the distribution of these scores were plotted. The box bounds represent the first and third quartiles and the black lines at the middle of the boxes show the medians. Top and/or bottom whiskers represent 1.5x of the range between the third and the first quartiles (interquartile range). Circles represent outliers. Two-sided Wilcoxon rank-sum test was performed. **c** Metaplot of the mean number of Sfpq CLIP peaks (±SEM) in region surrounding cryptic last exons or constitutive last exons of the same gene (control). **d**, **e** CLIP-seq peak distribution (top) and RNA-seq coverage plots (mid) from representative CLE-containing introns. The *Y*-axis scale of the read coverage plots is optimized for CLE/intronic reads. Sequence homology of flanking exons from orthologous CLE-containing genes is compared and the relative position of zebrafish CLEs is shown at the bottom.
**f** Heatmap illustrating increased inclusion (ΔPSI) of 68 CLEs upregulated in ALS-mutant background at different stages of neuronal differentiation. ΔPSI values of non-significant events are set to 0. Induced pluripotent stem cells (iPSC); neural precursors (NPC); "patterned" precursor motor neurons (ventral spinal cord; pMN); post-mitotic but electrophysiologically immature motor neurons (MN); electrophysiologically mature MNs (mMN). **g** Representative RNA-seq coverage plots from ALS-derived iPSC dataset of CLEs upregulated in VCP^*mu* samples.

transcriptional effects. Premature termination has been shown to act as a means of transcriptional control[61], and while the exact relationship between the use of CLEs and the expression of full-length transcripts depends on the gene (Fig. 1f, g and Supplementary Fig. 1c), at least a subset of CLEs appear to form at the expense of longer isoforms. Once transcribed, CLE transcripts with their differing 3′-terminal sequences may show altered patterns of stability, subcellular localization, or translation. Indeed, changes to last exon usage have been shown to be particularly important in localizing transcripts to particular neuronal compartments[9]. Finally, once translated, some short peptides may have novel or dominant negative functions compared to the full-length protein products.

We previously showed that loss of *sfpq* leads to an array of morphological and neurodevelopmental abnormalities in zebrafish embryos, including loss of brain boundaries and altered motor axon morphology[26]. However, the mechanism by which those abnormalities formed was unresolved. Here, we found that CLEs contribute to at least one aspect of the *sfpq* phenotype: the dominant negative *epha4b* truncated transcript induces hindbrain boundary defects. Moreover, a subset of the CLE-terminated short transcripts identified in *sfpq*[−/−] is predicted to affect axon growth and connectivity.

While CLE formation is clearly detectable under pathological conditions of loss of *sfpq*, our data do not preclude the possibility of CLEs being expressed under non-pathological conditions. Although the CLEs are not annotated in the current zebrafish, mouse, and human genomes, it is possible that they may be regulated in a spatio-temporal manner such that they only appear in select tissues or cell types at specific developmental time points. Indeed, this possibility is supported by the relatively low expression of SFPQ in non-neuronal tissue[25,26], and by low-level detection of the *epha4b* CLE transcript in siblings by RT-PCR (Supplementary Fig. 5c). Early termination of long pre-mRNAs has been shown to be a developmentally controlled regulatory mechanism: the RNA-binding protein Sex-lethal promotes the formation of truncated transcripts during short nuclear cycles in *Drosophila*[62], and downregulation of the cleavage and polyadenylation factor PCF11 during differentiation of mouse C2C12 myoblast cells suppresses intronic polyadenylation to promote expression of long genes[63]. Further examination of CLE expression in wild-ype animals across development may identify possible role of these truncated transcripts in normal tissues.

**Cryptic exons in neurodegenerative disease.** Neurodegenerative diseases such as Alzheimer's, ALS, and FTD are frequently characterized by altered localization and function of splicing factors[32,64–66]. The ALS-associated proteins transactivation response element DNA-binding protein 43 (TDP-43) and fused in sarcoma (FUS) regulate alternative splicing and alternative polyadenylation[67–72]. TDP-43 has been shown to act as a repressor of cryptic exons, a minority of which contain polyadenylation sites and thus would form CLEs[72]. Stathmin-2 is one of the latter and rescue of its normal full-length expression in TDP-43-knockdown cell culture improves axonal growth in this model[70,71], indicating that CLEs are pathogenic across various splicing protein-dependent pathologies. These findings place CLEs at the center of priority for understanding molecular mechanisms of neurodegenerative diseases and developing new ways to diagnose and treat these increasingly prevalent disorders.

## Methods

**Zebrafish husbandry.** Zebrafish (*Danio rerio*) were reared in accordance with the Animals (Scientific Procedures) Act 1986 under CH Home Office Project License 70/7577. Fish were maintained on a 14-h light/10-h dark cycle at 28 °C. Embryos were cultured in fish water containing 0.01% methylene blue to prevent fungal growth. Wild-type fish were AB strain from the Zebrafish International Resource Center (ZIRC), while *sfpq* null mutants were *sfpq*[kg4126].

**RNA-seq and 3′ mRNA-seq.** RNA was extracted from 24 hpf *sfpq*[−/−] embryos and their heterozygous or homozygous wild-type siblings using the RNeasy Mini Kit (Qiagen). Enriched mRNA was sequenced using the Illumina HiSeq 2500 with 50 bp paired-end reads. Libraries for 3′ mRNA-seq was prepared and sequenced by Lexogen using using QuantSeq 3′ mRNA-Seq Library Prep Kit FWD for Illumina.

**3′ RACE.** RNA was extracted from 24 hpf *sfpq*[−/−] embryos using the RNeasy Mini Kit (Qiagen). Reverse transcription was performed using the 3′ RACE System for Rapid Amplification of cDNA Ends (ThermoFisher). cDNA was amplified in two subsequent PCR reactions, using the adapter primer as a reverse primer and the primers listed in Supplementary Data 9 as forward primers. After the second amplification step, PCR products were gel-purified and sequenced directly.

**CLIP-qPCR.** Dechorionated 24 hpf wild-type fish were irradiated (twice at 0.8 J/cm², 254 nm) and deyolked using high calcium Ringer's solution (116 mM NaCl, 2.9 mM KCl, 10 mM CaCl2, 5 mM HEPES, pH 7.2) with 0.3 mM PMSF and 1 mM EDTA. After several washes, embryos were lysed using PXL buffer (0.1% SDS, 0.5% deoxycholate, 0.5% NP-40) and homogenized using a plastic pestle. Lysates were treated with 10 µL diluted RNAseI (1:500 dilution; Thermo Fisher) and 2 µL Turbo DNase (Thermo Fisher) at 37 °C for 3 min on a shaking incubator. Protein-RNA complexes were purified by centrifugation and 5% of the lysate was retained as input. The remaining lysate was split into two and its volume was topped up to 100 µL using PXL buffer. In all, 100 µL of protein A Dynabeads (Thermo Fisher) primed with 10 µg of either anti-SFPQ antibody (ab38148) or anti-IgG antibody (MA5-14453) were added to each lysate and incubated at 4 °C for an hour on a rotator. Bound SFPQ-RNA complexes were purified and washed thrice in high salt wash buffer (50 mM Tris-HCL, pH 7.4, 1 M NaCl, 1 mM EDTA, 1% Igepal, 0.1% SDS, 0.5% sodium deoxycholate). Subsequently, bound complexes were washed twice in PNK wash buffer (20 mM Tris-HCL, pH 7.4, 10 mM MgCl2, 0.2% Tween-20) and followed by proteinase K digestion (Thermo Fisher). Bound RNAs were purified using phenol–chloroform extraction followed by reverse transcription to generate cDNAs. Relative amounts of SFPQ-bound RNAs was quantified by qPCR using primers: (Supplementary Data 9).

**CRISPR/Cas9.** gRNAs were formed from chemically synthesized Alt-R®-modified crRNAs from Integrated DNA Technologies (IDT). Each crRNA was suspended in duplex buffer to 100 µM concentration, then a crRNA:tracrRNA duplex was formed by combining 3 µl crRNA, 3 µl 100 µM tracrRNA, and 19 µl duplex buffer at 95 °C for five minutes, then cooled to room temperature and stored at −20 °C. To make gRNA:Cas9 RNP complexes, a mix was formed as follows: 1.5 µl each gRNA, 0.75 µl 2 M KCl, 1.25 µl EnGen Spy Cas9 NLS (NEB). The mix was incubated at 37 °C for 5 min, then brought to room temperature. One nanoliter of the gRNA:Cas9 complex was injected into embryos at the 1-cell stage. The gRNAs used are listed in Supplementary Data 9.

**RNA and morpholino injections.** The *epha4b* cryptic transcript was amplified from cDNA and inserted into the multi-cloning site of plasmid pCS2 + (Addgene). To make the early stop-containing transcript, site-directed mutagenesis was performed using Quikchange Site-Directed Mutagenesis Kit (Agilent). In-vitro transcription reaction was performed on linearized plasmid using the mMessage mMachine SP6 Transcription Kit (ThermoFisher), and the RNA was purified using a Mini Quick Spin Column (Roche). In all, 100 pg RNA was injected into the embryo at the one-cell stage.

For morpholino knockdown of the *epha4b* cryptic exon, embryos were injected into the yolk at the one-cell stage with 0.1 pmol of Epha4b splice junction morpholino or mismatch (Supplementary Data 9).

**In situ hybridization.** Linearized plasmids containing the antisense sequence for *rfng*[73], *deltaA*[74], or the *epha4b* cryptic exon were transcribed into RNA probes using DIG labeling mix (Roche) according to the manufacturer's instructions. Probes were purified using Mini Quick Spin Columns (Roche). In situ hybridization reaction was performed by fixing embryos in 4% PFA overnight, followed by washing in PBST and methanol. Embryos were then rehydrated in PBST and treated with 1 µl of 10 mg/ml Proteinase K (Sigma) in 1000 ml PBST for 15 min. Embryos were then fixed again in 4% PFA for 20 min and pre-blocked in Hyb mix (1% Boehringer Block, 50% formamide, 4X SSC, 1 mg/ml torula mRNA, 0.1% CHAPS, 0.1% Tween-20, 5 mM EDTA, 100 µg/ml Heparin) for 4–6 h at 65 °C. In total, 25 µl probe was diluted in 300 µl Hyb mix and incubated overnight with embryos at 65 °C. Embryos were then washed in Hyb mix, followed by 2x SSC/CHAPS and 0.2x SSC/CHAPS. This was followed by a wash in MAB buffer (150 mM NaCL, 100 mM Maleic Acid, 0.1% Tween-20) and incubation in MAB/blocking solution (2% Roche Blocking Solution in 1x MAB), both at room temperature. Anti-DIG-AP antibody (Roche) was added 1:4000 overnight at 4 °C. Embryos were washed in MAB for a full day, then developed using NTMT (50 mM

MgCl₂, 100 mM NaCl, 100 mM Tris-HCl pH 9.5, 0.1% Tween-20) supplemented with NBT/BCIP (Roche).

**qPCR**. RNA was extracted from 24 to 28 hpf *sfpq−/−* embryos and heterozygous or WT siblings using the RNease Mini Kit (Qiagen). Embryos were identified based on phenotype, and 30 embryos were pooled together for each RNA extraction. In all, 1 μg of extracted RNA was used in a reverse transcriptase reaction using the Superscript III First Strand cDNA Synthesis Kit (Invitrogen). In all, 250 ng of cDNA was used in qPCR reactions with the LightCycler 480 SYBR Green I Master Mix (Roche). Each sample was compared against a *B-actin* control reaction.

**Bioinformatics**. For expression and splicing analyses of 24 hpf *sfpq−/−* RNA-seq data using Cufflinks package[40], reads were mapped to zebrafish GRCz9 assembly and differential expression analysis were carried out using default settings.

For expression and splicing analyses of 24 hpf *sfpq−/−* RNA-seq data using Whippet pipeline[41], a GRCz10 Ensembl-based index was generated using Whippet's index building function from the Ensembl-based fasta (ftp://ftp.ensembl.org/pub/release-91/fasta/danio_rerio/dna/Danio_rerio.GRCz10.dna.toplevel.fa.gz) and gene annotation files (ftp://ftp.ensembl.org/pub/release-91/gtf/danio_rerio/Danio_rerio.GRCz10.91.gtf.gz). Quantification of aligned RNA-seq reads were done as follows:

whippet-quant.jl read1.fastq.gz read2.fastq.gz --biascorrect -x index.jls
-o <output_directory > --sam <SAM_output_directory>

The above quantification function outputs gene and isoform level read count matrices. Differential gene or isoform expression analyses were identified using the edgeR package with the estimateGLMRobustDisp function[75]. Differential splicing events were identified using Whippet's delta analysis function with default parameters. An event with a Probability score exceeding 80% is classified as significantly regulated. Type of splicing event for each alternative exon was determined from Whippet's output file under column Type. Alternative exons which are not currently annotated in the GRCz10 assembly are classified as "cryptic" using custom R-scripts.

To determine if CLE-containing transcripts are polyadenylated, 3′ mRNA-seq data were trimmed to remove adapter contamination, polyA read-through and low-quality tails using BBDuk (https://jgi.doe.gov/data-and-tools/bbtools/bb-tools-user-guide/bbduk-guide/):

bbduk.sh in = input_1.fastq out = output_1.fastq ref = <TruSeq RNA adapter sequence>

k = 13 ktrim = r useshortkmers = t mink = 5 qtrim = r trimq = 10 minlength = 20

Trimmed reads were aligned to GRCz10 genome using HISAT2 (https://www.nature.com/articles/s41587-019-0201-4) with spliced alignment disabled:

hisat2 -p <num_threads > --no-spliced-alignment -x<GRCz10 index>
-U trimmed_read1.fastq -S output.sam

Reads mapping upstream of internal (A) homopolymers (10 consecutive adenosines with one allowed mismatch within a 20-nt window) were removed using custom scripts. The last nucleotide of the remaining reads mapping to known genes was annotated as a cleavage site (CS). Clusters of CSs were then generated by merging CSs which are within ≤10 nt of each other across all experimental samples. CS clusters were further refined by keeping clusters containing ≥3 reads in at least one sample as well as having a consensus PAS hexamer (ATTAAA or AATAAA) within a 50-nt window (40 nt upstream and 10 nt downstream) of the middle of CS clusters.

CS clusters were next assigned to CLE-containing genes (Supplementary Fig. 2a). Clusters are assigned to cryptic last exons if located between the start of CLE and 3′ end of its downstream intron. Clusters belong to control last exons if found within the last constitutive exon of its gene or within 50-nt downstream padding. Quantification of relative cleavage/polyadenylation efficiency (RCE) for each last exon was calculated as

$$RCE = \frac{N_e}{\sum_{i=0}^{n} N_i} \qquad (1)$$

where $N_e$ is the total number of reads matching cleavage/polyadenylation clusters belonging to exon $e$, and $n$ is the total number of reads mapping to cleavage/polyadenylation clusters in the same gene. Statistical significance changes in cleavage efficiency was determined using two-tailed Fisher's exact test by comparing $N_e$ and $(\sum_{i=0}^{n} N_i - N_k)$ values between experiments. FDR was calculated using Benjamini-Hochberg correction method. Normalized change in RCE statistic between experimental (e) and control (c) samples were calculated as

$$\Delta RCE_{norm} = \frac{RCE_e - RCE_c}{RCE_e + RCE_c} \qquad (2)$$

For analyses of conditional Sfpq knock-out mouse model[23] dataset, the above Whippet pipeline was carried out using Ensembl's GRCm38 fasta (ftp://ftp.ensembl.org/pub/release-99/fasta/mus_musculus/dna/Mus_musculus.GRCm38.dna.toplevel.fa.gz) and annotation (ftp://ftp.ensembl.org/pub/release-99/gtf/mus_musculus/Mus_musculus.GRCm38.99.gtf.gz) files. For analyses of conditional ALS-derived iPSC differentiation dataset[31], the above Whippet pipeline was carried out using Ensembl's GRCh37 fasta (ftp://ftp.ensembl.org/pub/grch37/current/fasta/homo_sapiens/dna/Homo_sapiens.GRCh37.dna.toplevel.fa.gz) and annotation

(ftp://ftp.ensembl.org/pub/release-75/gtf/homo_sapiens/Homo_sapiens.GRCh37.75.gtf.gz) files.

To construct CLE-containing transcripts, read alignments from Whippet were sorted, indexed and assembled using the StringTie program[76]. Ensembl's GRCz10 transcriptome was used as reference and assembly was done for each biological replicate as follows:

stringtie <file1.bam>-p <num_threads>-o<file1.gtf>- G<reference>

Assembled transcripts from each sample were subsequently combined using StringTie's merge function using GRCz10 annotations as reference. CLE-containing isoforms were identified by intersecting exon coordinates from the merged transcript assembly with CLE coordinates from Whippet delta analysis output. Intersection operation was done in R using Bioconductor's GenomicRanges package[77]. Analyses on the coding potential of CLE isoforms and its functional loss of protein domains were carried out using custom R-scripts.

For the analyses of introns from which the CLEs were spliced from, intronic features were extracted from the custom-assembled transcript in R using Bioconductor's GenomicFeatures package[77]. A list of the largest, non-overlapping introns was generated and annotated for an overlap with a CLE segment using GenomicRanges' reduce and subsetByOverlaps functions respectively. The relative position of CLEs within its intron was determined using psetdiff operation followed by extracting the width of the upstream intronic segment. To determine if CLEs are spliced from very large introns by chance, we applied weights to each intron found in CLE-containing genes proportional to its size and sampled, with replacement, one intron per gene for 10 or 100 iterations using the sample() function in R.

For the analyses of CLE conservation, 8-way PhastCons and PhyloP score data were downloaded from UCSC (http://hgdownload.soe.ucsc.edu/goldenPath/danRer7/phastCons8way/fish.phastCons8way.bw and http://hgdownload.soe.ucsc.edu/goldenPath/danRer7/phyloP8way/fish.phyloP8way.bw). Coordinates of CLE-containing introns were converted to GRCv9 using UCSC's LiftOver function and binned into 1 kb sequence using a sliding window technique (1 bp steps). Average PhastCons score of each bin was calculated using bedtools' "map" function and bins overlapping CLE were annotated through intersection. Conservation scores of each CLE and 250 nt of its flanking introns were calculated using the same tool. To refine the conservation regions surrounding the intron-CLE borders, average PhastCons score were calculated for 10 nt windows including 30 nt of each exonic ends.

For the analyses of SFPQ-binding motifs within sequences surrounding CLEs, its Position-Specific Scoring Matrix was downloaded from RBPmap (http://rbpmap.technion.ac.il/download.html) and manually converted into a MEME motif format (http://meme-suite.org/doc/meme-format.html). Occurrence of SPFQ binding sites was analyzed using MEME's FIMO program (http://meme-suite.org/doc/fimo.html) using the following parameters:

fimo --thresh 0.005 --o <output_directory><SFPQ_PSSM><CLE_fasta>

The average number of SFPQ-binding motifs were calculated for 25 nt windows of flanking intronic sequence including 25 nt of each exonic ends.

For the analyses of acceptor splice site strengths, 23 nt sequence of the intron-exon border (20 nt intron and 3 nt exon) of CLEs or control last exons were generated using custom R scripts. Scores for each sequence were predicted using MaxEntScan model (Yeo and Burge, https://pubmed.ncbi.nlm.nih.gov/15285897/). Sequence logo depicting the consensus 3′ splice site sequence was plotted using WebLogo (https://weblogo.berkeley.edu/logo.cgi)

**Reporting summary**. Further information on research design is available in the Nature Research Reporting Summary linked to this article.

## Data availability
The RNA-seq data analyzed in Figs. 1 and 3 is available at ArrayExpress, accession number E-MTAB-9113. The 3′ mRNA-seq data analyzed in Fig. 2 is available at ArrayExpress, accession number E-MTAB-9899. The data supporting the findings of this study are available from the corresponding authors upon reasonable request. Source data are provided with this paper.

## Code availability
Computer code used in this study is described in the Methods section.

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

## Acknowledgements

This work was supported by the Biotechnology and Biological Sciences Research Council (BB/P001599/1 to C.H. and BB/R001049/1 to E.V.M.); the European Commission (Project ID 734791; E.V.M); the Wellcome Trust (Tech Dev. WT093389 and Equipment WT094819 to C.H.) and EMBO (fellowship ALTF 1530-2015 to P.M.G.). We thank Richard Poole for help with the TopHat pipeline.

## Author contributions

P.M.G. designed and conducted the experiments, analyzed the data, and wrote the paper. F.H. designed and conducted the experiments, performed computational studies, analyzed the data, and wrote the paper. E.V.M. designed experiments, analyzed the data, and wrote the paper. C.H. designed experiments, analyzed the data, and wrote the paper.

## Competing interests

The authors declare no competing interests.
