## [Peer Review File · Nature Communications]

REVIEWER COMMENTS

Reviewer #1 (Remarks to the Author):

In this manuscript, Gordon et al., report that loss of *sfpq* results in activation of a few hundreds of cryptic last exons in zebrafish, mouse and human. The activation of these new isoforms can reduce the expression of the full-length transcript (demonstrated for one example) or their associated proteins can act as dominant negatives (demonstrated for another example). While previous work had documented links between splicing factors (or spliceosome components) and early transcription termination, Gordon et al. presents here a novel phenomenon in which early termination arises from splicing in of cryptic exons. These are important results and their biological importance for neurodevelopment is nicely shown by the CLE-containing *epha4b* mRNA isoform in zebrafish.

The less satisfying aspect of the manuscript is the lack of information regarding the mechanism and evidence that this regulatory system is biologically different from other premature transcription termination mechanisms described previously. I would expect further progress along these lines for publication in Nature Communication.

Main comments:

1. While both Cufflinks and Whippet detect changes in the usage of skipped exons from RNA-seq data, they don't have any specific algorithm to identify (and quantify) the usage of alternative last exons. Therefore, I suggest to use a more specialized pipeline to identify (and quantify) more accurately the usage of last exons and alternative polyadenylation sites from RNA-seq data, such as APATrap (Ye et al., 2018) or mountainClimber (Cass et al., 2019).

Most importantly, RNA-seq data is far from optimal to detect last exon usage due to a decreased coverage at the end of transcripts ("edge effect"). In my opinion, it is necessary to use a specialized method to detect alternative polyadenylation sites in a genome-wide scale (e.g., polyA-seq) to accurately identify CLEs. I was glad to see that the authors validated the usage of alternative polyadenylation sites with 3'RACE in Figure 2g. However, it is not clear how many clones they sequenced for each gene and, in any case, they only validated the usage of CLEs in 6 genes. Also, since neither Cufflinks nor Whippet are specialized pipelines for identification of polyadenylation sites and the results obtained with both methods are quite different, a more conservative approach would be to use only those 25 genes with CLEs identified with both pipelines for all downstream analyses, instead of using all genes identified by Whippet.

2. From my perspective, the phenomenon described here is very similar to what some authors call "telescripting", in which the binding of U1 SNRNP suppress the usage of cryptic and premature polyadenylation sites. The only clear difference is that the polyadenylation sites are located in introns in telescripting and in exons in SFPQ-mediated CLE. The concept of "telescripting" should be introduced in the introduction and the authors should incorporate deeper analyses showing the difference. The first time "telescripting" is mentioned in the text is in the discussion section (page 12) and the authors highlight that unlike "telescripting", CLEs are not biased toward the 5' end of genes. However, they show in Figure 5b that CLEs do have a significant bias towards the 5' end in human genes. As a side note, the literature cited for "telescripting" is somehow arbitrary. They shouldn't exclude crucial papers as Kaida et al., 2010 and Almada et al., 2013. For the analysis on human genes, could the authors show that CLEs and "telescripting" affect different subsets of genes? With specific pipelines/methods to identify polyadenylation sites, could the authors analyze the usage of cryptic and premature polyadenylation sites (located in introns vs exons) in *sfpq*^{-/-} embryos compared to their siblings?

3. There is no real mechanistic analysis in this manuscript. The CLIP experiment validated the binding of SFPQ in only three genes. What happens in the majority of genes? And even in those three genes

where SFPQ binds directly, can the authors hypothesize a molecular mechanism?

It could be worth to analyze the RNA binding protein knock-down data from the ENCODE project to search for other factors involved in this phenomenon in human cells. Knock-down of other splicing factors could yield a similar result or even rescue the *sfpq*^{-/-} phenotype.

Furthermore, it is demonstrated here that CLEs can dampen the expression of the full-length transcripts only in the case of *b4galt2* (but not for the other genes tested). An even for *b4galt2* the effect seems quite small. In this regard, it will be better to move Figure 4a to supplementary material.

Specific comments:

1. In Figure 1c, how are these categories analyzed? Please explain further. In Figure 1d, cryptic exons are not annotated as exons at all or are they not annotated as last exons? Many last exons are annotated as internal exons in the current genome annotation.

2. In Figure 1f, there are 103 CLE genes while in the text the number seems to be 106 (line 104).

3. It was recently showed that internal skipped exons can impact the usage of alternative promoters and activate cryptic promoters nearby (Fiszbein et al., 2019). Have the authors checked if the activation of CLEs have an impact in the usage of alternative/cryptic promoters? It might be interesting specially for *epha4b* where two isoforms are expressed (Figure 1e right panel).

4. It is interesting that CLE exons are evolutionarily conserved. Could the authors analyze the strength of their splice sites with MaxEntScan (Yeo et al., 2003) or any other method?

5. For the analysis of the *sfpq* phenotype, instead of only focusing on *epha4b*, could the authors do a genome-wide analysis looking at possible impacts of many protein functions? Table S4 shows an attempt to look at all possible impacts, but is not clearly explained in the text.

6. In relation to the "telescripting" phenomenon in which premature and cryptic intronic polyadenylation sites are used, why do the authors hypothesize that in this case the inclusion of the CLE is needed? This becomes very clear in the *epha4b* example when the MO rescues the phenotype. The polyadenylation site is only used when the exon is included?

7. Is the MO affecting the expression of the *epha4b* gene? The rescued phenotype is stronger in the MO-transfected embryos compared to their wild-type siblings (Figure 4e,d)

8. The conservation across vertebrates of the repression of CLEs by SFPQ is a very interesting issue, but the analysis presented here is a bit confusing and might not be deep enough. From the text it seems that, instead of analyzing the mouse data as they did for zebrafish, the authors only look at the mouse ortholog genes of their CLE-identified zebrafish genes. That approach is very limited. And as a result, they only identified CLE formation in 4 mouse genes. However, in Table S5 there is a longer list of CLEs from mouse. Thus, it is not clear what they did and what the results are. For human iPSCs derived from ALS patients, the analysis seems to be more extended, but again, it is not clear what those 76 CLE events up-regulated in ALS mutant backgrounds represent. A deeper and better explained analysis is needed.

Reviewer #2 (Remarks to the Author):

In this work by Gordon and colleagues the authors investigate the mechanism of action for the conserved RNA binding protein SFPQ. Beginning with analysis of the transcriptome of zebrafish embryos mutant for *sfpq*, the authors identify a previously unappreciated class of RNAs harboring cryptic last exons (CLEs). Using bioinformatic analyses, they map the CLEs to long introns of genes with important functions in the nervous system. They show through in situ hybridization and qPCR assays that these CLE variants are expressed in mutants and are minimally or not detected in normal embryos. Using overexpression and knockdown assays the authors show that expression of CLEs containing variants of Notch pathway components can perturb gene expression and hindbrain development, while knocking down these genes with splice blocking agents, and thus preventing production of the CLE variant can restore some aspects of gene expression and patterning in the *sfpq* mutant hindbrain. Finally, they extend their finding to mammalian systems and iPSC-derived neurons and show through bioinformatics analyses that CLE variants are conserved and that CLE containing transcripts are detected in iPSCs derived from ALS patients, raising the exciting possibility that these truncated transcripts contribute to pathology in human neurodegenerative disease.

The authors use multiple strategies to test the function of CLE transcripts, including overexpression and knock-down approaches and they examine the expression of a couple of different markers as readout for suppression in assays where they test whether expression of CLE accounts for disruptive phenotypes in the *sfpq* mutant brain. Overall the data are clearly presented and compelling and will be of interest to the field as the study provides mechanistic into *sfpq* function and reveals a previously unappreciated mode of regulation. I have only minor comments to clarify some of the experimental details in the methods and to include the numbers of embryos examined for some of the experiments.

- 1) In Figure 3 panels b-d more description is needed for the graphs – for example it does not state in the panel or legend what the numbers 1-10 above the qPCR data represent. Are these alternating grey and white numbered areas representative of exons/introns, do they represent the 50bp bins defined in panel A, or are they otherwise related to the predicted binding sites above? Please clarify.
- 2) On Pg. 8, the authors state that they “performed RT-qPCR on pooled injected *sfpq*^{-/-} embryos to measure the expression levels of the normal *b4galt2* transcripts and saw a significant rescue of the longer transcripts compared to the uninjected *sfpq*^{-/-} control”. In the methods, they state the “RNA was extracted from 24-28 hpf *sfpq* embryos”, but a bit more detail is needed for this experiment. For example, how were mutants identified? Were they selected based on phenotype or were they genotyped and then pooled? How many embryos were in each pool?
- 3) In Fig 4C the example showing the effective phenocopy of the *sfpq* mutant hindbrain rhombomere boundary defect when *epha4b* CLE overexpression is striking and quite compelling, but the manuscript does not state how many embryos were examined and show this pattern. These numbers should be included in the main text or the figure legend.
- 4) Fig. 4d *rfng* staining: states quantified for “3 lateral view samples per each condition”. Although based on the representative shown the mutant phenotype appears to be suppressed it is important to include the total number examined and fraction with this phenotype even if the intensity of expression of additional individuals was not quantified in the same manner.
- 5) On page 5 the authors report down regulation of CLE containing genes. Can the authors elaborate on this “downregulation effect” in the discussion? Was this a change in the overall abundance of the transcripts for the genes examined or rather a shift in the type of RNA produced with no change in expression levels if the various types of transcripts were summed? If there is a change in abundance overall, do the authors think this is an NMD mechanism or a more targeted effect/gene specific regulation associate with CLEs, either transcriptional versus post transcriptional?
- 6) The gene IDs are cut off in some of the supplemental pdf files; however, the original excel files are provided so this is not really an issue.

Reviewer #3 (Remarks to the Author):

In this manuscript, Gordon et al. reported an interesting molecular phenotype of SFPQ knockout in zebrafish embryos: the activation/upregulation of RNA isoforms utilizing a new last exon (cryptic last exons, CLEs) mostly reside in large introns in ~100 genes. The authors focused on several examples and showed that some of the CLEs are likely generated via simultaneous activation of cryptic 3' splice sites and polyadenylation sites in introns. In one case the authors showed that the activation of CLE may down-regulate the full-length mRNA. In another example the author showed overexpressing a CLE isoform is detrimental for neural development in zebrafish. The authors also analyzed RNA-seq data from mouse SFPQ knockout and human ALS iPSC cells with low SFPQ and identified ~100 putative CLEs in each case. The author concluded that SFPQ has a conserved role in repressing pathogenic CLEs.

The strength of the manuscript is in the observation of SFPQ depletion activating cryptic last exons in large introns throughout the gene. This is distinct from known mechanisms such as U1 snRNP depletion activating intronic polyadenylation in 5' introns without activating cryptic 3' splice sites. However, more evidence is needed to show those events are indeed processed from alternative splicing and polyadenylation. Many conclusions are based on individual examples instead of systematic analysis. The evidence presented are often insufficient to rule out alternative hypothesis. In many cases the authors need to be more specific and tune down their interpretation of the data.

Major comments:

1. Annotation of CLEs. By definition, CLEs are generated by activation of both a cryptic 3' splice site in intron (defining the 5' end of the cryptic exon) and a poly (A) site further downstream in the intron (defining the 3' end of the exon). Further evidence are required to support that the ~100 events in each of the species are authentic CLEs. For example, although the two CLEs in Fig 1e showed well-defined boundaries with splicing junction reads, the two mouse CLEs in fig 5a lack those features and could well be intronic polyadenylation events without splicing, or transcripts initiated within introns, or just variation of intronic reads in the library. This is especially an issue considering no replicates were used to define those rare/noisy events, and that CLEs called by two tools (cufflinks / Whippet) showed a very small overlap (25 of 106, fig s1b) (The authors should not call such small number of occurrence as "pervasive"). Both of the zebrafish and mouse RNA-seq data appear to be sequencing of total RNA rather than poly(A) selected RNA, thus there is no evidence on whether polyadenylation is happening for most CLEs. Although the authors validated six examples with 3' RACE (Fig 2g), these examples can be biased (e.g. picked from events with clear 3' end boundary from RNA-seq data). It is recommended that the authors perform 3'-end poly(A) RNA sequencing to validate the polyadenylation status of those RNAs. If it is impossible to carry out additional experiments due to COVID-19, the authors can examine published 3'-end sequencing data from zebrafish, mouse, and human cells to see if evidence can be found supporting intronic polyadenylation downstream of the cryptic 3' splice site. It is also recommended that the authors verify that the cryptic splice sites are authentic in that 1) there are sufficient number of junction reads and 2) the splice site is canonical AG acceptor site.

2. Long intron preference. The authors claimed in the abstract that "these CLEs appear preferentially in long introns of genes". This seems to be referring to the observation that most introns with CLEs are large (fig 2b and fig 5b). However, this bias towards large introns is expected if CLEs simply occur randomly in the genome: most of them will occur in large introns because these introns occupy larger sequence space. In this case, mechanistically there is no preference for large introns or bias against small introns. To see if there is a preference for large introns beyond what is expected from a random occurrence model, the authors can compare to a background distribution of CLE intron size generated by randomly assigning CLE events to introns with a probability proportional to intron size.

3. Impact of CLE on gene expression. The authors claimed in the abstract that CLEs "dampen gene expression output". This is assuming the biogenesis of the long and short isoforms are mutually

exclusive. The strongest evidence was shown by the example of *b4galt2*, for which the increase of CLE is associated with a decrease of the full length mRNA, and CRISPR-mediated deletion of the CLE leads to an increase of the full length mRNA. However, the rescue is minor and rescue was not observed in two other genes tested. The other evidence used to support the claim was the observation that ~40% (42 of 103, the author incorrectly claimed more than half) CLE-containing genes are down-regulated (fig 1f). However, the fact that 96% (1,173) of the down-regulated genes do not have CLE suggests one should not assume CLE is causing the down-regulation. Instead, most likely the 42 CLE-containing genes are down-regulated by a yet to be defined mechanism that affects most of the >1000 genes. The authors did not comment on how would SFPQ knockout downregulate the large number of genes. One possibility is transcriptional regulation, as SFPQ has been reported to interact with other DNA-binding proteins and regulate transcription. In fact, the mouse study the authors cited showed that SFPQ co-transcriptionally associated with polymerase and regulates the transcription of long genes. It will be interesting to see if those 1215 down-regulated genes are enriched for long genes. If so, the authors should consider a model in which SFPQ facilitates transcription elongation, and in the knockout mutant a slow polymerase resulted in the utilization of cryptic splice sites and polyadenylation sites that randomly occur in large introns. At the minimal the authors should tune down the claim that CLEs dampens gene expression.

4. Peptide encoded by CLE isoforms. The authors also claimed in the abstract that CLEs, in particular the CLE in *epha4b* gene, give rise to short peptide that interfere with normal gene function. However, the authors did not show evidence that a peptide is generated, and did not exclude a model in which the RNA causes the phenotype. The putative peptide would end with a C-terminal encoded by intronic sequence, which is likely unfolded and will be degraded. One experiment that can help address this is to inject the *epha4b* CLE isoform with a mutated start codon (i.e. replace the control in fig 4c).

5. SFPQ binding to CLE flanking sequences. The authors performed CLIP-qPCR (incorrectly claimed CLIP-seq is done in discussion) and found evidence of SFPQ binding near the CLE in three genes (Fig 3). However, there doesn't seem to be a strong association between binding and putative motifs. Further, the mouse study the authors cited included a CLIP-seq experiment, but the authors did not perform a meta-gene analysis similar to fig 3a to see if indeed SFPQ tends to bind near the CLE events. The examples shown in fig 5a do not support such a conclusion. The authors are recommended to sequence the CLIP samples.

Minor comments:

The introduction started with the importance of compartment and localization in neurons but throughout the manuscript no discussion on whether the observation is related to protein localization. Related to this, Taliaferro et al. reported previously (<https://doi.org/10.1016/j.molcel.2016.01.020>) that alternative last exon usage affects RNA localization in neurons. The authors should discuss the relevance. If possible, check the overlap of the alternative last exons.

Fig 1a/b: how many of the down-regulated genes are identified by both methods? Are any of those involved in splicing and polyadenylation that may explain CLE events?

Would be better to show a scatter plot to see how strong is the negative correlation between fold changes of CLE and gene expression.

Four genes (*Epha4b*, *Cpped1*, *Fam172a*, *Exoc4*) were mentioned to contain CLEs in both zebrafish and mouse. Are these the only genes conserved? Among the 144 CLEs unregulated in mouse, what fraction are also among in the ~100 CLEs found in zebrafish? if found in both organisms, do they tend to be found after the same exon?

Fig s2b: difficult to assess if this is significant because most of the difference seems to come from those with a score of 0. Phastcons scores do not have the resolution outside highly conserved regions.

The authors are recommended to use phyloP scores instead.

The 3' splice site signal from fig 2f may come from a small number of unannotated coding exons, or recently lost coding exons. Would be better to know how many CLE events show this pattern of increased conservation at the 3' splice site.

As a general audience without a zebrafish research background, it is unclear to me how significant is the difference shown in fig 4 and how important is the loss of those patterns.

Figure S3 is missing.

Reviewed by Xuebing Wu. Relevant expertise: RNA and genomics, especially splicing and polyadenylation in mammalian cells.

Response to reviewers

We are delighted that all three reviewers appear to have read our manuscript with great interest and agree that it uncovers a novel level of regulation of gene expression. They also raised some questions and requested several additional experiments. We apologize for the delay in providing these additional data, imposed by the pandemic. Our point-by-point response to reviewers' comments is as follows.

Reviewer #1

"Gordon et al. presents here a novel phenomenon in which early termination arises from splicing in of cryptic exons. These are important results and their biological importance for neurodevelopment is nicely shown"

The reviewer wants to see more information on the mechanism regulating cryptic last exons, more specifically evidence showing that these exons differ from premature transcription termination events described previously.

1. *While both Cufflinks and Whippet detect changes in the usage of skipped exons from RNA-seq data, they don't have any specific algorithm to identify (and quantify) the usage of alternative last exons. Therefore, I suggest to use a more specialized pipeline to identify (and quantify) more accurately the usage of last exons and alternative polyadenylation sites from RNA-seq data, such as APATrap (Ye et al., 2018) or mountainClimber (Cass et al., 2019).*

RNA-seq data is far from optimal to detect last exon usage due to a decreased coverage at the end of transcripts ("edge effect"). In my opinion, it is necessary to use a specialized method to detect alternative polyadenylation sites in a genome-wide scale (e.g., polyA-seq) to accurately identify CLEs. I was glad to see that the authors validated the usage of alternative polyadenylation sites with 3'RACE in Figure 2g. However, it is not clear how many clones they sequenced for each gene and, in any case, they only validated the usage of CLEs in 6 genes. Since neither Cufflinks nor Whippet are specialized pipelines for identification of polyadenylation sites, ... a more conservative approach would be to use only those 25 genes with CLEs identified with both pipelines for all downstream analyses, instead of using all genes identified by Whippet.

We agree that a more specialized method is needed to verify the results from our standard RNA-seq analysis. We performed 3' mRNA-seq, with results that confirm our original Cufflinks and Whippet analyses (see new Figure 2 and Figure S2). In addition, we have expanded our 3'RACE results to 10 genes and clarified in Methods that 3' RACE sequencing was done directly on PCR products and not on plasmid clones.

2. *From my perspective, the phenomenon described here is very similar to what some authors call "telescripting", in which the binding of U1 SNRNP suppress the usage of cryptic and premature polyadenylation sites. The only clear difference is that the polyadenylation sites are located in introns in telescripting and in exons in SFPQ-mediated CLE. The concept of "telescripting" should be introduced in the introduction and the authors should incorporate deeper analyses showing the difference. [...] the authors highlight that unlike "telescripting", CLEs are not biased toward the 5'end of genes. However, they show in Figure 5b that CLEs do have a significant bias towards the 5'end in human genes.*

We thank the reviewer for suggesting additional papers regarding telescripting. We have added those to our discussion, and introduced telescripting early on in the revised manuscript. We have expanded our discussion showing the mechanistic differences between CLEs and telescripting. Crucially, telescripting is splicing-independent repression of cryptic polyadenylation sites by U1 snRNP. In this case, loss of U1 binding to pre-mRNA leads to premature cleavage/polyadenylation without activating an intervening 3' splice site. We do not see this type of splicing-independent events in our dataset. Moreover, we see a broad range of CLE locations (especially in

zebrafish and mouse genes) in contrast with telescripting events, which are often biased to transcription start-proximal positions.

*For the analysis on human genes, could the authors show that CLEs and “telescripting” affect different subsets of genes? With specific pipelines/methods to identify polyadenylation sites, could the authors analyze the usage of cryptic and premature polyadenylation sites (located in introns vs exons) in *sfpq*^{-/-} embryos compared to their siblings?*

We have included data from our paper (in submission, available on BioRxiv) showing that CLEs are not enriched in mutants of zebrafish *snRNP70*, a U1 snRNP component required for telescripting.

*3. There is no real mechanistic analysis in this manuscript. The CLIP experiment validated the binding of SFPQ in only three genes. What happens in the majority of genes? And even in those three genes where SFPQ binds directly, can the authors hypothesize a molecular mechanism? It could be worth to analyze the RNA binding protein knock-down data from the ENCODE project to search for other factors involved in this phenomenon in human cells. Knock-down of other splicing factors could yield a similar result or even rescue the *sfpq*^{-/-} phenotype. Furthermore, it is demonstrated here that CLEs can dampen the expression of the full-length transcripts only in the case of *b4galt2* (but not for the other genes tested). And even for *b4galt2* the effect seems quite small. In this regard, it will be better to move Figure 4a to supplementary material.*

We believe that the reviewer has underestimated the importance of the *b4galt2* functional results. Our analyses show that expression of full-length transcripts is dampened for half of the genes generating CLEs in *sfpq* mutants. We functionally assessed whether inhibiting CLE usage would restore expression of the normal transcripts for three randomly picked genes affected and got a rescue for one. This rescue is partial because we are measuring changes in genetic mosaics, where CLE is inactivated only in a subset of the cells. These results support our conclusion that CLEs can function by either dampening gene expression or generating dominant negative products. We therefore hope that the reviewer will agree on our decision to keep this Figure in the main paper.

Elucidation of further mechanistic details that allow SFPQ to block CLEs would require a separate line of experiments going beyond the scope of our current study. We have attempted to genetically delete SFPQ binding sites using genome editing but this has proved challenging (possibly due to repetitive nature of intronic sequences). We have expanded our discussion on the possible mechanism. Direct repression of CLEs by SFPQ is consistent with our CLIP experiments and with previous work on SFPQ showing its direct interaction with Pol II to promote transcriptional elongation.

Given the impact that CLEs have on the *sfpq* phenotype and the connection between CLEs and neurodegenerative diseases, we would strongly argue that the CLE phenomenon is of great importance that warrants publication in Nature Communications.

Minor points

1. In Figure 1c, how are these categories analyzed? Please explain further. In Figure 1d, cryptic exons are not annotated as exons at all or are they not annotated as last exons? Many last exons are annotated as internal exons in the current genome annotation.

We added a description of the source of splicing categories to the Methods section and defined cryptic splicing events in the text.

2. In Figure 1f, there are 103 CLE genes while in the text the number seems to be 106 (line 104).

We fixed this in the text

3. It was recently showed that internal skipped exons can impact the usage of alternative promoters and activate cryptic promoters nearby (Fiszbein et al., 2019). Have the authors checked if the activation of CLEs have an impact in the usage of alternative/cryptic promoters? It might be interesting specially for *epha4b* where two isoforms are expressed (Figure 1e right panel).

We performed this analysis for *epha4b* and found no difference in the use of alternative promoters. See attached figure R1.

4. It is interesting that CLE exons are evolutionarily conserved. Could the authors analyze the strength of their splice sites with MaxEntScan (Yeo et al., 2003) or any other method?

We performed this analysis and found that 3'ss strength on CLEs is slightly lower than on control last exons but still containing a canonical U2AF binding motif. See figures S3c-d.

5. For the analysis of the *sfpq* phenotype, instead of only focusing on *epha4b*, could the authors do a genome-wide analysis looking at possible impacts of many protein functions? Table S4 shows an attempt to look at all possible impacts, but is not clearly explained in the text.

We believe this refers to Table S6 instead of Table S4. We added reference to Table S6 in the Results section.

6. In relation to the “telescripting” phenomenon in which premature and cryptic intronic polyadenylation sites are used, why do the authors hypothesize that in this case the inclusion of the CLE is needed? This becomes very clear in the *epha4b* example when the MO rescues the phenotype. The polyadenylation site is only used when the exon is included?

The CLE phenomenon is distinct from telescripting in that telescripting does not require splicing but CLEs do. This is shown clearly by the *epha4b* MO rescue, which indicates that in the absence of a splice acceptor site (3' ss), the cryptic exon is unable to form and the cryptic transcript is not expressed, therefore not using the cryptic polyA site.

7. Is the MO affecting the expression of the *epha4b* gene? The rescued phenotype is stronger in the MO-transfected embryos compared to their wild-type siblings (Figure 4e,d)

We have clarified this in the text: *in-situ* staining is always more intense in morphants, suspected to be due to increase stability of mRNAs by MOs. For this reason, the best comparison is to mismatch control instead of to uninjected siblings.

8. The conservation across vertebrates of the repression of CLEs by SFPQ is a very interesting issue, but the analysis presented here is a bit confusing and might not be deep enough. From the text it seems that, instead of analyzing the mouse data as they did for zebrafish, the authors only look at the mouse ortholog genes of their CLE-identified zebrafish genes. That approach is very limited. And as a result, they only identified CLE formation in 4 mouse genes. However, in Table S5 there is a longer list of CLEs from mouse. Thus, it is not clear what they did and what the results are. For human iPSCs derived from ALS patients, the analysis seems to be more extended, but again, it is not clear what those 76 CLE events up-regulated in ALS mutant backgrounds represent. A deeper and better explained analysis is needed.

We apologize for the confusion. We did not only look for mouse orthologs of CLE-identified zebrafish genes, but instead did a whole-genome analysis. As we have now explained in the text, we would not expect to find much overlap between our zebrafish results and the human or mouse results as the overall pool of transcribed genes is very different between the three conditions. However, we did see some

overlap, the four genes that we mentioned in the text. We have added details to the coverage plots and added a better description in the Results section.

Reviewer #2

*“Overall the data are clearly presented and compelling and will be of interest to the field as the study provides mechanistic into *sfpq* function and reveals a previously unappreciated mode of regulation. I have only minor comments to clarify some of the experimental details in the methods and to include the numbers of embryos examined for some of the experiments.”*

1) In Figure 3 panels b-d more description is needed for the graphs – for example it does not state in the panel or legend what the numbers 1-10 above the qPCR data represent. Are these alternating grey and white numbered areas representative of exons/introns, do they represent the 50bp bins defined in panel A, or are they otherwise related to the predicted binding sites above? Please clarify.

We have added scales on the figure to clarify that each area represents 100 nt. We have also edited the figure legend accordingly.

2) On Pg. 8, the authors state that they “performed RT-qPCR on pooled injected *sfpq*^{-/-} embryos to measure the expression levels of the normal *b4galt2* transcripts and saw a significant rescue of the longer transcripts compared to the uninjected *sfpq*^{-/-} control”. In the methods, they state the “RNA was extracted from 24-28 hpf *sfpq* embryos”, but a bit more detail is needed for this experiment. For example, how were mutants identified? Were they selected based on phenotype or were they genotyped and then pooled? How many embryos were in each pool?

The Methods section was expanded to describe how embryos were identified, collected, and pooled prior to RNA extraction.

3) In Fig 4C the example showing the effective phenocopy of the *sfpq* mutant hindbrain rhombomere boundary defect when *epha4b* CLE overexpression is striking and quite compelling, but the manuscript does not state how many embryos were examined and show this pattern. These numbers should be included in the main text or the figure legend.

We have included these numbers in the figure legend.

4) Fig. 4d *rftng* staining: states quantified for “3 lateral view samples per each condition”. Although based on the representative shown the mutant phenotype appears to be suppressed it is important to include the total number examined and fraction with this phenotype even if the intensity of expression of additional individuals was not quantified in the same manner.

- We have included these numbers in the figure legend.

5) The authors report down regulation of CLE containing genes. Can the authors elaborate on this “downregulation effect” in the discussion? Was this a change in the overall abundance of the transcripts for the genes examined or rather a shift in the type of RNA produced with no change in expression levels if the various types of transcripts were summed? If there is a change in abundance overall, do the authors think this is an NMD mechanism or a more targeted effect/gene specific regulation associate with CLEs, either transcriptional versus post transcriptional?

We have expanded on this in the Results and Discussion: downregulation indicates overall change in abundance of the transcripts, although the CLE-containing transcripts are enriched. We are thus seeing both change in overall expression and a shift in the mRNAs produced.

6) The gene IDs are cut off in some of the supplemental pdf files; however, the original excel files are provided so this is not really an issue.

We have fixed this.

Reviewer #3

“The strength of the manuscript is in the observation of SFPQ depletion activating cryptic last exons in large introns throughout the gene. This is distinct from known mechanisms such as U1 snRNP depletion activating intronic polyadenylation in 5’ introns without activating cryptic 3’ splice sites. However, more evidence is needed to show those events are indeed processed from alternative splicing and polyadenylation. Many conclusions are based on individual examples instead of systematic analysis. The evidence presented are often insufficient to rule out alternative hypothesis.”

Major comments:

1. *Annotation of CLEs. By definition, CLEs are generated by activation of both a cryptic 3’ splice site in intron (defining the 5’ end of the cryptic exon) and a poly (A) site further downstream in the intron (defining the 3’ end of the exon). Further evidence are required to support that the ~100 events in each of the species are authentic CLEs. [...] Although the authors validated six examples with 3’ RACE (Fig 2g), these examples can be biased (e.g. picked from events with clear 3’ end boundary from RNA-seq data). It is recommended that the authors perform 3’-end poly(A) RNA sequencing to validate the polyadenylation status of those RNAs. If it is impossible to carry out additional experiments due to COVID-19, the authors can examine published 3’-end sequencing data from zebrafish, mouse, and human cells to see if evidence can be found supporting intronic polyadenylation downstream of the cryptic 3’ splice site. It is also recommended that the authors verify that the cryptic splice sites are authentic in that 1) there are sufficient number of junction reads and 2) the splice site is canonical AG acceptor site.*

We have performed 3’ mRNA-seq as well as additional 3’RACE experiments. Please see our response to Reviewer #1, main comment 1. In addition, we have performed splice site analysis using MaxEntScan (Figure S3c-d) and found clear U2AF binding sites at the expected positions.

2. *Long intron preference. The authors claimed in the abstract that “these CLEs appear preferentially in long introns of genes”. This seems to be referring to the observation that most introns with CLEs are large (fig 2b and fig 5b). However, this bias towards large introns is expected if CLEs simply occur randomly in the genome: most of them will occur in large introns because these introns occupy larger sequence space. In this case, mechanistically there is no preference for large introns or bias against small introns. To see if there is a preference for large introns beyond what is expected from a random occurrence model, the authors can compare to a background distribution of CLE intron size generated by randomly assigning CLE events to introns with a probability proportional to intron size.*

We performed this statistical analysis and found that CLE introns are longer than would be expected by chance just based on larger sequence space of long introns. See figure S3a.

3. *Impact of CLE on gene expression. The authors claimed in the abstract that CLEs “dampen gene expression output”. This is assuming the biogenesis of the long and short isoforms are mutually exclusive. The strongest evidence was shown by the example of b4galt2, for which the increase of CLE is associated with a decrease of the full length mRNA, and CRISPR-mediated deletion of the CLE leads to an increase of the full length mRNA. However, the rescue is minor and rescue was not observed in two other genes tested. The other evidence used to support the claim was the observation that ~40% (42 of 103, the author incorrectly claimed more than half) CLE-containing genes are down-regulated (fig 1f). However, the fact that 96% (1,173) of the down-regulated genes do not have CLE suggests one should not assume CLE is causing the down-regulation. Instead, most likely the 42 CLE-containing genes are down-regulated by a yet to be defined mechanism that affects most of the >1000 genes.*

We indeed found 1173 transcripts underrepresented in the mutant. As the RNAseq is done at 28hpf, these can be both primary and secondary consequences of the loss of SFPQ. Loss of transcripts is definitely the more common change in expression. It could be caused by degradation of mis-spliced transcripts or by loss of transcription of the genes as SFPQ is also involved in transcriptional events. However, what we noticed regarding the CLEs is that the exons upstream of CLE are read in similar level as in the siblings while the downstream exons are downregulated, strongly suggesting that the production of CLE-containing shorter transcripts prevent the normal expression of the longer normal variants.

We functionally assessed whether inhibiting CLE usage would restore expression of the normal transcripts for three randomly picked affected genes and got a rescue for one. This rescue is not spectacular because we are using a transient approach therefore measuring changes in genetic mosaics, where deletion of the CLE is happening only in a fraction of the cells in these embryos. Considering this mosaicism, the effect measured is rather substantial.

It will be interesting to see if those 1215 down-regulated genes are enriched for long genes. If so, the authors should consider a model in which SFPQ facilitates transcription elongation, and in the knockout mutant a slow polymerase resulted in the utilization of cryptic splice sites and polyadenylation sites that randomly occur in large introns. At the minimal the authors should tune down the claim that CLEs dampens gene expression.

The under-expressed genes are not enriched for long transcripts. And most of the 1173 genes are not producing CLEs

4. Peptide encoded by CLE isoforms. The authors also claimed in the abstract that CLEs, in particular the CLE in *epha4b* gene, give rise to short peptide that interfere with normal gene function. However, the authors did not show evidence that a peptide is generated, and did not exclude a model in which the RNA causes the phenotype. The putative peptide would end with a C-terminal encoded by intronic sequence, which is likely unfolded and will be degraded. One experiment that can help address this is to inject the *epha4b* CLE isoform with a mutated start codon (i.e. replace the control in fig 4c).

We performed that experiment, injecting the *epha4b* cryptic transcript with an early stop codon and found it to be no different from uninjected control. See Figure 5b.

5. SFPQ binding to CLE flanking sequences. The authors performed CLIP-qPCR (incorrectly claimed CLIP-seq is done in discussion) and found evidence of SFPQ binding near the CLE in three genes (Fig 3). However, there doesn't seem to be a strong association between binding and putative motifs. Further, the mouse study the authors cited included a CLIP-seq experiment, but the authors did not perform a meta-gene analysis similar to fig 3a to see if indeed SFPQ tends to bind near the CLE events. The examples shown in fig 5a do not support such a conclusion. The authors are recommended to sequence the CLIP samples.

We and others have attempted to perform CLIP-seq on SFPQ but it has never worked in our system due in part to the relatively weak performance of SFPQ antibodies in zebrafish. We have examined the mouse data to look for enrichment of SFPQ binding near CLEs, and we indeed found binding peaks near CLEs. See Figure 6c.

Minor comments:

The introduction started with the importance of compartment and localization in neurons but throughout the manuscript no discussion on whether the observation is related to protein localization. Related to this, Taliaferro et al. reported previously (<https://doi.org/10.1016/j.molcel.2016.01.020>) that alternative last exon usage affects RNA localization in neurons. The authors should discuss the relevance. If possible, check the overlap of the alternative last exons.

We have added consideration of localization to the discussion. Directly comparing the last exon usage between our dataset and the Taliaferro paper would not be expected to find much overlap because the overall pool of transcribed genes in mouse neuronal cell lines is very different from those found in developing zebrafish embryos.

Fig 1a/b: how many of the down-regulated genes are identified by both methods? Are any of those involved in splicing and polyadenylation that may explain CLE events?

We have made a list of the commonly down-regulated genes; see enclosed Table R1. Splicing and polyadenylation-related genes are annotated, and there is no enrichment.

Would be better to show a scatter plot to see how strong is the negative correlation between fold changes of CLE and gene expression.

We have included this scatter plot in Figure S1c.

Four genes (Epha4b, Cpped1, Fam172a, Exoc4) were mentioned to contain CLEs in both zebrafish and mouse. Are these the only genes conserved? Among the 144 CLEs unregulated in mouse, what fraction are also among in the ~100 CLEs found in zebrafish? if found in both organisms, do they tend to be found after the same exon?

We now discuss in the text why we expect to see a low overlap: given that our RNA-seq and the mouse RNA-seq were performed in different tissues and developmental stages, the overall pool of transcribed genes is different. For the four genes that are conserved, we have improved our coverage plots in Figure 6 to show that the flanking exons are highly conserved. We have added info on this to the Results section.

Fig s2b: difficult to assess if this is significant because most of the difference seems to come from those with a score of 0. Phastcons scores do not have the resolution outside highly conserved regions. The authors are recommended to use phyloP scores instead.

We have re-analyzed the data with Phastcons and show the results of these new analyses in Figures 3f and S3e-g.

The 3' splice site signal from fig 2f may come from a small number of unannotated coding exons, or recently lost coding exons. Would be better to know how many CLE events show this pattern of increased conservation at the 3' splice site.

The 3' splice site was highly conserved in about 10% of CLEs: see Figure S3j.

As a general audience without a zebrafish research background, it is unclear to me how significant is the difference shown in fig 4 and how important is the loss of those patterns.

We have clarified this in the text. The loss of these patterns is of interest not only because it indicates a large change in hindbrain boundary formation, but also because it provides an explanation for the results from our previous paper, when we described the boundary defect but had not found a means by which that defect occurred.

Figure S3 is missing.

We have connected supplementary figures to main text figures. There was no Figure S3 (which now would be S4 as we have added a new Figure 2) because the associated main figure does not have any supplemental information.

REVIEWERS' COMMENTS

Reviewer #1 (Remarks to the Author):

I appreciate the authors' efforts to address the issues related to last exon identification and telescripting of my previous report, as well as the other points raised. The manuscript is significantly stronger and I support its publication.

Just a few minor comments in case them help:

- I still have some questions about the definition of cryptic splicing events. Can the authors confirm that cryptic exons are not annotated as exons at all? For example, Cufflinks might treat some last exons as non-annotated when they are indeed annotated but classified as internal and not as last exons.
- It is very nice that the authors included data from their other paper showing that CLEs are not enriched in mutants of zebrafish snRNP70. Is it possible to include a longer section about it (including supplementary figures) in this paper as well? I think it is crucial to support the conclusions of this paper.
- In terms of the usage of alternative promoters (minor comment 3), I was thinking if the relative usage of the alternative promoters of epha4b (or annotated alternative first exons) change, not the absolute values of the predominant promoter.

Reviewer #2 (Remarks to the Author):

In this work by Gordon and colleagues the authors investigate the mechanism of action for the conserved RNA binding protein SFPQ. Beginning with analysis of the transcriptome of zebrafish embryos mutant for *sfpq*, the authors identify a previously unappreciated class of RNAs harboring cryptic last exons (CLEs). Using bioinformatic analyses, they map the CLEs to long introns of genes with important functions in the nervous system. They show through in situ hybridization and qPCR assays that these CLE variants are expressed in mutants and are minimally or not detected in normal embryos. Using overexpression and knockdown assays the authors show that expression of CLEs containing variants of Notch pathway components can perturb gene expression and hindbrain development, while knocking down these genes with splice blocking agents, and thus preventing production of the CLE variant can restore some aspects of gene expression and patterning in the *sfpq* mutant hindbrain. Finally, they extend their finding to mammalian systems and iPSC-derived neurons and show through bioinformatics analyses that CLE variants are conserved and that CLE containing transcripts are detected in iPSCs derived from ALS patients, raising the exciting possibility that these truncated transcripts contribute to pathology in human neurodegenerative disease.

In this revised version, the authors have thoughtfully addressed the concerns raised in the previous version of this manuscript by adding additional details, clarifications, new data and new analyses. These additions provide further support for the authors conclusions and strengthen the manuscript. The writing is clear, the data are well presented and the findings are exciting.

Reviewer #3 (Remarks to the Author):

The authors have addressed my previous comments.

Xuebing Wu

Response to reviewers

We are delighted with the support given by all three reviewers. Below are our responses to the minor comments raised.

Reviewer #1 (Remarks to the Author):

"I still have some questions about the definition of cryptic splicing events. Can the authors confirm that cryptic exons are not annotated as exons at all? For example, Cufflinks might treat some last exons as non-annotated when they are indeed annotated but classified as internal and not as last exons."

The exons detected in our study were defined as cryptic if it is non-existent in the zebrafish GRCz10 reference annotation. The mouse orthologues of a few of these genes also contain CLEs not annotated in the GRCm38 transcriptome (Figure 6d-e and Supplementary Figure 6d-e)

"It is very nice that the authors included data from their other paper showing that CLEs are not enriched in mutants of zebrafish snRNP70. Is it possible to include a longer section about it (including supplementary figures) in this paper as well? I think it is crucial to support the conclusions of this paper."

We have now extended the section discussing lack of CLEs in the snRNP70 null mutant. We additionally include here Supplementary Figure R2 that extend this analysis; however, these figures are much more relevant to and appropriate for the Nikolau et al. study than this paper, so they should not be included in this manuscript.

"In terms of the usage of alternative promoters (minor comment 3), I was thinking if the relative usage of the alternative promoters of epha4b (or annotated alternative first exons) change, not the absolute values of the predominant promoter."

The Percent Spliced In (PSI) index shown in Supplementary Figure R1 measures relative usage of the alternative promoters, where the sum of PSI_{upstream} and $PSI_{\text{downstream}}$ in each sample equates to 100%. The change in PSI between *sfpq* and siblings remain not statistically significant when either PSI index were used.

Supplementary Figure R1: Appearance of CLE in *sfpq*^{-/-} have no effect on alternative promoter usage.

Usage (shown as Percent Spliced In) of the alternative downstream promoter of *epha4b* gene. Data was obtained from Whippet splicing analysis and mean PSI from 3 biological replicates is shown (\pm SD).

Supplementary Figure R2: SFPQ-regulated CLEs are not co-regulated by SNRNP70

- (a) Breakdown of types of splicing events regulated by SFPQ and SNRNP70
- (b) Only a small fraction of SNRNP70-regulated events are cryptic
- (c) SNRNP70 do not co-regulate SFPQ-regulated CLEs

Reviewer #2 (Remarks to the Author):

“In this work by Gordon and colleagues the authors investigate the mechanism of action for the conserved RNA binding protein SFPQ. Beginning with analysis of the transcriptome of zebrafish embryos mutant for sfpq, the authors identify a previously unappreciated class of RNAs harboring cryptic last exons (CLEs). Using bioinformatic analyses, they map the CLEs to long introns of genes with important functions in the nervous system. They show through in situ hybridization and qPCR assays that these CLE variants are expressed in mutants and are minimally or not detected in normal embryos. Using overexpression and knockdown assays the authors show that expression of CLEs containing variants of Notch pathway components can perturb gene expression and hindbrain development, while knocking down these genes with splice blocking agents, and thus preventing production of the CLE variant can restore some aspects of gene expression and patterning in the sfpq mutant hindbrain. Finally, they extend their finding to mammalian systems and iPSC-derived neurons and show through bioinformatics analyses that CLE variants are conserved and that CLE containing transcripts are detected in iPSCs derived from ALS patients, raising the exciting possibility that these truncated transcripts contribute to pathology in human neurodegenerative disease.

In this revised version, the authors have thoughtfully addressed the concerns raised in the previous version of this manuscript by adding additional details, clarifications, new data and new analyses. These additions provide further support for the authors conclusions and strengthen the manuscript. The writing is clear, the data are well presented and the findings are exciting.”
We thank you for your comments and support in our study.

Reviewer #3 (Remarks to the Author):

“The authors have addressed my previous comments.”
Thank you for taking time in reviewing our revised manuscript.